# Arp2/3 complex-driven spatial patterning of the BCR enhances immune synapse formation, BCR signaling and B cell activation

Madison Bolger-Munro[1,2]*, Kate Choi[1,2], Joshua M Scurll[3], Libin Abraham[1,2,3], Rhys S Chappell[3], Duke Sheen[1,2], May Dang-Lawson[1,2], Xufeng Wu[4], John J Priatel[5,6], Daniel Coombs[3], John A Hammer[4], Michael R Gold[1,2]*

[1]Department of Microbiology and Immunology, University of British Columbia, Vancouver, Canada; [2]Life Sciences Institute, I3 Research Group, University of British Columbia, Vancouver, Canada; [3]Department of Mathematics, Institute of Applied Mathematics, University of British Columbia, Vancouver, Canada; [4]Cell Biology and Physiology Center, National Heart, Lung and Blood Institute, National Institutes of Health, Bethesda, United States; [5]Department of Pathology and Laboratory Medicine, University of British Columbia, Vancouver, Canada; [6]BC Children's Hospital Research Institute, Vancouver, Canada

**Abstract** When B cells encounter antigens on the surface of an antigen-presenting cell (APC), B cell receptors (BCRs) are gathered into microclusters that recruit signaling enzymes. These microclusters then move centripetally and coalesce into the central supramolecular activation cluster of an immune synapse. The mechanisms controlling BCR organization during immune synapse formation, and how this impacts BCR signaling, are not fully understood. We show that this coalescence of BCR microclusters depends on the actin-related protein 2/3 (Arp2/3) complex, which nucleates branched actin networks. Moreover, in murine B cells, this dynamic spatial reorganization of BCR microclusters amplifies proximal BCR signaling reactions and enhances the ability of membrane-associated antigens to induce transcriptional responses and proliferation. Our finding that Arp2/3 complex activity is important for B cell responses to spatially restricted membrane-bound antigens, but not for soluble antigens, highlights a critical role for Arp2/3 complex-dependent actin remodeling in B cell responses to APC-bound antigens.
DOI: https://doi.org/10.7554/eLife.44574.001

*For correspondence:
mbolgerm@mail.ubc.ca (MB-M);
mgold@mail.ubc.ca (MRG)

**Competing interests:** The authors declare that no competing interests exist.

## Introduction

The binding of membrane-bound ligands to their receptors can initiate multiscale spatial reorganization of receptors and their signal transduction machinery. This presumably optimizes receptor signaling and is often driven by remodeling of the cortical cytoskeleton. The formation of an immune synapse between a lymphocyte and an antigen-presenting cell (APC) is an excellent model for studying the dynamic changes in receptor and cytoskeleton organization that occur during cell-cell interactions and how they impact cellular activation (*Dustin and Groves, 2012*; *Harwood and Batista, 2010*; *Harwood and Batista, 2011*; *Song et al., 2014*). B lymphocytes play a critical role in health and disease by producing antibodies and cytokines. Although B cells can be activated by the binding of soluble antigens (Ags) to the B cell receptor (BCR), APCs that capture and concentrate Ags are potent activators of B cells (*Batista and Harwood, 2009*).

The immune synapse is a transient and highly dynamic structure that is characterized by a rapidly evolving reorganization of plasma membrane components and signaling proteins. When B cells encounter Ag on the surface of an APC, BCRs undergo rapid actin-dependent spatial reorganization that amplifies BCR signaling (*Harwood and Batista, 2010*) and reduces the amount of Ag required for B cell activation (*Batista et al., 2001*; *Depoil et al., 2008*; *Weber et al., 2008*). Initial BCR signaling induces localized actin severing and transiently uncouples the cortical cytoskeleton from the plasma membrane (*Freeman et al., 2011*; *Gupta et al., 2006*; *Treanor et al., 2011*). These changes allow for increased mobility of BCRs within the plasma membrane and enable the formation of BCR microclusters that are initially distributed throughout the B cell-APC contact site (*Batista et al., 2001*; *Depoil et al., 2008*). Because BCR signaling output depends on BCR-BCR interactions, microcluster formation amplifies BCR signaling. BCR clustering leads to phosphorylation of the CD79a/b (Ig-$\alpha$/$\beta$) signaling subunit of the BCR (*Abraham et al., 2016*; *Dal Porto et al., 2004*). This allows recruitment of the Syk tyrosine kinase to the BCR, which nucleates the formation of a 'microsignalosome', a microcluster-based signaling complex that includes the Btk tyrosine kinase, phospholipase C$\gamma$2 (PLC$\gamma$2), and phosphoinositide 3-kinase (PI3K) (*Treanor et al., 2009*; *Weber et al., 2008*). BCR signaling stimulates actin polymerization at the periphery of the B cell-APC contact site (*Fleire et al., 2006*), which exerts outward force on the plasma membrane to generate membrane protrusions (*Mogilner and Oster, 1996*). This allows the B cell to scan the APC surface for additional Ag, resulting in further BCR microcluster formation and enhancement of BCR signaling (*Fleire et al., 2006*; *Weber et al., 2008*).

Subsequent retraction of the B cell membrane is accompanied by the centripetal movement of BCR-Ag microclusters, which coalesce to form the characteristic central supramolecular activation complex (cSMAC) of an immune synapse (*Fleire et al., 2006*). BCR-mediated extraction of Ag from the APC membrane, which allows B cells to present Ags and elicit T cell help, occurs at the cSMAC in naive B cells (*Nowosad et al., 2016*; *Yuseff et al., 2013*). In T cells, centripetal (i.e. retrograde) actin flow is a driving force for T cell receptor (TCR) microcluster centralization and cSMAC formation (*Babich et al., 2012*; *Hammer and Burkhardt, 2013*; *Yi et al., 2012*). As new actin is polymerized at the cell periphery, the elastic resistance of the plasma membrane causes the peripheral actin network to flow toward the center of the synapse (*Ponti et al., 2004*). Actin dynamics are required for cSMAC formation in B cells (*Liu et al., 2012*; *Treanor et al., 2011*) but a role for actin retrograde flow has not been established.

Although the actin-dependent formation, movement, and spatial organization of BCR micoclusters are critical for APC-induced B cell activation, the molecular mechanisms that orchestrate this process are not fully understood. Actin organization and dynamics are controlled by a large network of actin-regulatory proteins (*Bezanilla et al., 2015*; *Chhabra and Higgs, 2007*; *Michelot and Drubin, 2011*; *Tolar, 2017*). There are two main modes of actin polymerization (*Pollard, 2007*). Formin-mediated linear actin polymerization, which is controlled by the RhoA GTPase, generates thin membrane protrusions such as filopodia (*Mattila and Lappalainen, 2008*). Alternatively, branched actin polymerization is mediated by the 7-subunit actin-related protein (Arp) 2/3 complex, which binds to existing actin filaments and nucleates new filaments that grow at a 70° angle from the mother filament (*Goley and Welch, 2006*). Receptor-induced activation of the Cdc42 and Rac GTPases causes conformational changes in nucleation-promoting factors (NPFs) belonging to the Wiskott-Aldrich Syndrome protein (WASp) family, which allows them to activate the Arp2/3 complex (*Rotty et al., 2013*). The Arp2/3 complex nucleates the branched actin networks that underlie the formation of sheet-like membrane protrusions such as lamellipodia (*Pollard and Borisy, 2003*). The peripheral actin network at the T cell immune synapse resembles that of the lamellipodia of migrating cells (*Dustin, 2007*; *Dustin and Cooper, 2000*).

A key role for the Arp2/3 complex in T and B cells is suggested by the dysregulated lymphocyte function in patients with Wiskott-Aldrich Syndrome (WAS). WAS is caused by mutations in either WASp or WASp-interacting protein (WIP) (*Candotti, 2018*) and is characterized by increased susceptibility to autoimmune disease, recurring infections, and predisposition to lymphomas and leukemias. In B cells, the WASp and N-WASP NPFs, as well as WIP, regulate actin dynamics and BCR signaling at the immune synapse (*Keppler et al., 2015*; *Liu et al., 2013*; *Liu et al., 2011*; *Westerberg et al., 2010*). Although WASp and N-WASP act upstream of the Arp2/3 complex, the role of Arp2/3 complex-dependent actin dynamics in B cell immune synapse formation and function has not been investigated.

B cell immune synapse formation was first demonstrated in the context of B cell-APC interactions (*Batista et al., 2001*). However, most subsequent studies have employed Ags that are attached to artificial planar lipid bilayers or to the cytoplasmic face of plasma membrane sheets in order to facilitate imaging of the contact site. We have developed systems for imaging APC-induced B cell immune synapse formation in real time (*Wang et al., 2018*) and now use this approach to investigate the role of Arp2/3 complex-dependent actin dynamics in B cell immune synapse formation and function. Here, we show that B cells interacting with live, intact APCs form BCR-Ag microclusters within dynamic actin-rich membrane protrusions that are dependent on the Arp2/3 complex. The coalescence of BCR-Ag microclusters into a cSMAC is also dependent on the dynamic branched actin networks that are nucleated by the Arp2/3 complex. Importantly, we show that Arp2/3 complex activity amplifies BCR signaling and is required for maximal B cell activation and proliferation.

## Results

### B cells generate dynamic actin-rich protrusions to scan for Ags on APCs

To visualize how B cells scan the surface of APCs to search for Ag, we generated APCs expressing a fluorescently tagged, transmembrane form of hen egg lysozyme (mHEL-HaloTag) and then imaged the B cell-APC contact site in real time (*Figure 1A*). When A20 murine B-lymphoma cells expressing the HEL-specific D1.3 transgenic BCR (A20 D1.3 B cells) were added to the APCs, they rapidly formed BCR-Ag microclusters throughout the contact site. Microclusters that formed at the periphery moved centripetally. Closer to the center of the contact site, microclusters coalesced with each other such that a cSMAC developed within ~10 min (*Figure 1B*). Although some BCR-Ag microclusters formed quite far from the center of the contact site (*Figure 1B* [yellow arrowheads]), nearly all the microclusters coalesced into the cSMAC after 10 min (*Figure 1B*, *Video 1*).

Membrane protrusions are generally driven by actin polymerization. To image how B cells probe the surface of APCs, we expressed the F-actin-binding protein F-Tractin-GFP in A20 D1.3 B cells and imaged their interaction with mHEL-HaloTag-expressing APCs using instant structured illumination microscopy (ISIM) (*York et al., 2013*) to obtain high spatiotemporal resolution. We found that B cells continually extended and retracted short-lived actin-rich protrusions across the surface of the APC (*Figure 1C* [colored arrowheads], *Figure 1D*, *Video 2*), resulting in an asymmetrical cell shape. This dynamic searching mechanism allowed A20 D1.3 B cells to scan a total area of ~100 $\mu m^2$ on the surface of the APC over 4 min of contact and gather Ag into microclusters (*Figure 1D*). BCR-Ag microclusters formed within actin-rich regions, including newly formed protrusions (*Figure 1C*, *Video 2*). These microclusters moved toward the center of the synapse along with the actin structures in which they were embedded, especially as membrane protrusions were retracted (*Figure 1E*, *Video 2*). As determined from the kymographs in *Figure 1E*, the velocities at which the peripheral F-actin structures and BCR-Ag microclusters moved toward the center of the synapse were similar, suggesting that actin dynamics drive the initial centripetal movement of these microclusters. At the same time, F-actin was cleared from the center of the contact site and BCR microclusters within this region coalesced with each other until a cSMAC was formed.

In contrast to their behavior on APCs, B cells exhibited largely radial spreading when they contacted immobilized anti-IgG antibodies (an Ag surrogate) on a rigid glass coverslip (*Figure 1F*, *Video 3*). The cells underwent continual outward spreading, reaching a contact area of ~150 $\mu m^2$ after 10 min (*Figure 1G*; see also Figure 3C,D). B cells spreading on immobilized anti-Ig formed multiple large lamellipodia, some of which persisted for up to 10 min (for example, *Figure 1F* [blue arrowheads]). They also formed short dynamic filopodia, many of which were retracted or engulfed by newly formed lamellipodia.

### The Arp2/3 complex is required for centralization of BCR-Ag microclusters

The Arp2/3 complex nucleates branched actin polymerization, which drives the formation of lamellipodial protrusions, resembling those that were observed in B cells spreading on APCs (*Figure 1C*). Because peripheral actin polymerization is associated with actin retrograde flow, we asked whether Arp2/3 complex activity is required for the centralization of BCR-Ag microclusters. To address this question, we used two complementary and well-characterized approaches to ablate the function of

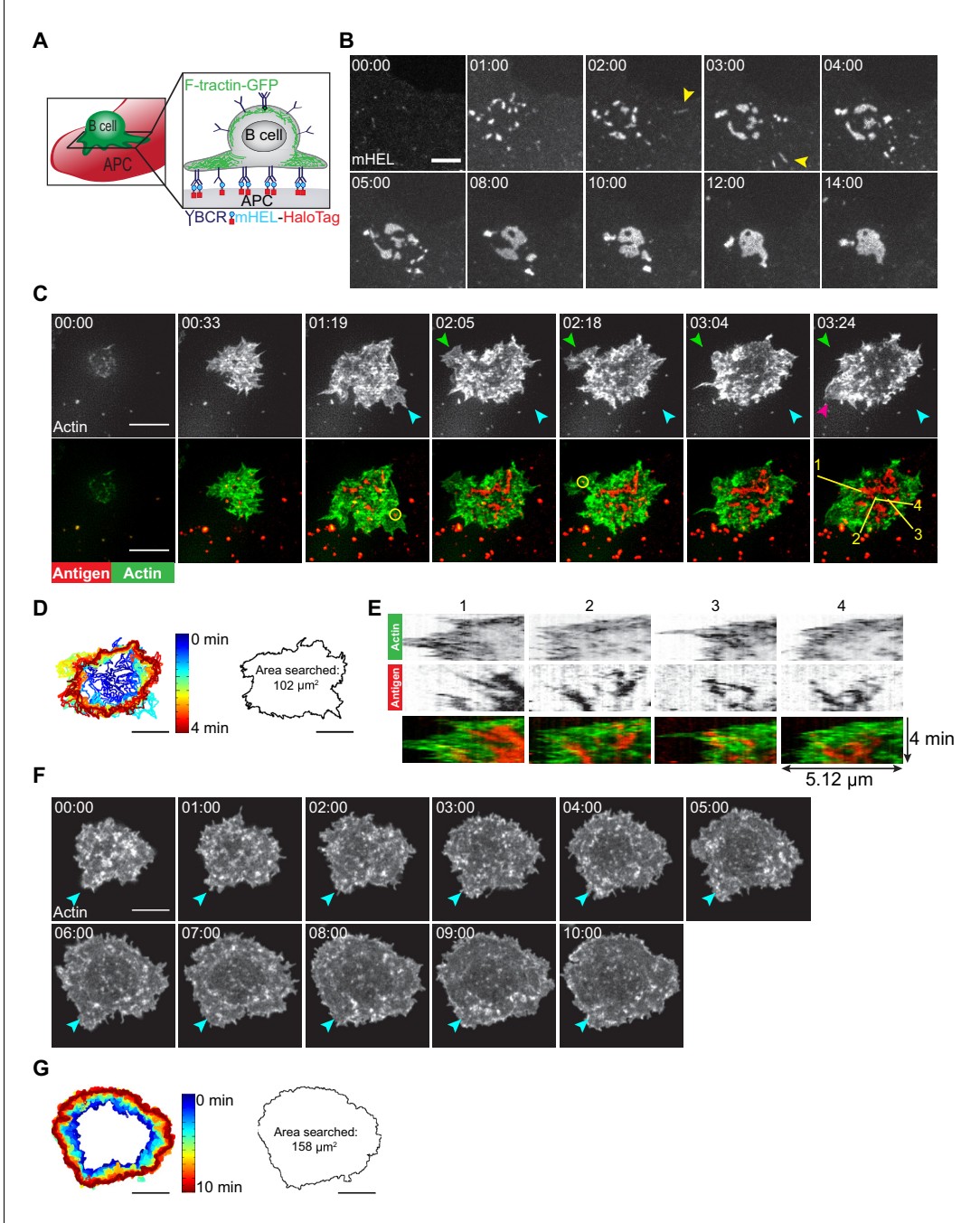

**Figure 1.** B cells generate dynamic actin-based probing protrusions in response to APC-bound Ags. (A) Schematic of the B cell-APC system. COS-7 cells expressing mHEL-HaloTag Ag act as APCs for B cells expressing HEL-specific BCRs. (B) A20 D1.3 B cells were added to COS-7 cells expressing mHEL-HaloTag and the B cell-APC contact site was imaged every 12 s for 14 min using spinning disk microscopy. Images from *Video 1* are shown. Yellow arrowheads indicate BCR-Ag microclusters that formed far from the center of the contact site. The data are representative of >40 cells from nine independent experiments. Times are shown in min:s. (C) A20 D1.3 B cells expressing F-Tractin-GFP (green) were added to COS-7 APCs expressing mHEL-HaloTag (red) and the B cell-APC contact site was imaged every 6.6 s for 4 min using ISIM. Images from *Video 2* are shown. The different colored arrowheads in the upper panels (actin channel) represent membrane protrusions that were extended or retracted. The yellow circles in the lower panels indicate BCR-Ag microclusters (red) that were formed on new actin protrusions. (D) For each frame of *Video 2*, the cell edge, as defined by F-actin, was overlaid as a temporally-coded time series (left). The total area searched by the B cell over the 4-min period of observation is shown on the right. (E) Kymographs showing the time evolution of fluorescence signals along the yellow lines in the lower right panel of (C) depict the centripetal movement of Ag clusters and the surrounding actin structures. The velocity of a particle is proportional to the angle at which its track deviates from a vertical line along the time axis. See *Figure 1—figure supplement 1* for additional explanation of kymograph analysis. (F) A20 B cells expressing
*Figure 1 continued on next page*

*Figure 1 continued*

F-Tractin-GFP were added to anti-IgG-coated coverslips and the contact site was imaged every 2 s for 10 min using a Zeiss Airyscan microscope. Images from *Video 3* are shown. Blue arrowheads represent a long-lived membrane protrusion. (G) A temporally coded time series representing the edge of the cell (left) was generated from *Video 3* and the total area searched by the B cell over the 10 min period of observation is shown (right). Scale bars: 5 μm.

DOI: https://doi.org/10.7554/eLife.44574.002

The following figure supplement is available for figure 1:

**Figure supplement 1.** Schematic of kymograph analysis.

DOI: https://doi.org/10.7554/eLife.44574.003

the Arp2/3 complex in B cells. First, we used siRNA to deplete Arp3 expression. Depletion of Arp3 was accompanied by reduced expression of Arp2 and p34, suggesting destabilization of the Arp2/3 complex (*Figure 2—figure supplement 1*), as observed previously (*Nicholson-Dykstra and Higgs, 2008*; *Zhang et al., 2017*). Second, we inhibited Arp2/3 complex activity using a pharmacological inhibitor, CK-666 (*Hetrick et al., 2013*). CK-666 locks the Arp2/3 complex in an open conformation, which prevents it from binding to existing actin filaments and nucleating the formation of new actin filaments. B cells treated with CK-666 did not exhibit increased cell death compared to controls (*Figure 2—figure supplement 2A,B*). Although the COS-7 APCs would also be exposed to CK-666 in these experiments, the CK-666 concentration that we used did not alter their F-actin organization or dynamics over a 30 min timescale (*Figure 2—figure supplement 3*).

We found that inhibiting or depleting the Arp2/3 complex greatly reduced the coalescence of BCR-Ag microclusters into a cSMAC (*Figure 2A,B*), even though there was no change in the total amount of Ag that was gathered into microclusters (*Figure 2—figure supplement 4A,B*) or the number of BCR-Ag microclusters that formed (*Figure 2—figure supplement 4C,D*). In A20 D1.3 B cells that were transfected with control siRNA (*Figure 2A*, *Video 4*) or treated with CK-689 (*Figure 2B*, *Video 6*), an inactive analog of CK-666, BCR-Ag microclusters exhibited obvious centripetal movement. By 10 min after contacting the APCs, ~60% of the B cells had gathered >90% of the Ag fluorescence into one or two large clusters at the center of the synapse, which we define as having formed a cSMAC (*Figure 2A,B*). By contrast, many of the BCR-Ag microclusters in the Arp3 siRNA-expressing cells (*Figure 2A*, *Video 5*) and the CK-666-treated cells (*Figure 2B*, *Video 7*) did not exhibit persistent centripetal motion and only ~15% of the cells in which the Arp2/3 complex was inhibited or depleted formed a cSMAC by 10 min (*Figure 2A,B*). Even after 30 min, very few CK-666-treated cells formed a cSMAC (data not shown).

The extent of BCR-Ag microcluster coalescence and cSMAC formation was further analyzed by quantifying the distance of each microcluster on an individual B cell from the center of mass of the Ag fluorescence (i.e. the center of the immune synapse). For each B cell, the average microcluster distance was calculated for each frame in the video and this value was normalized to the maximum average distance for that cell (*Figure 2—figure supplement 4E,F*). In A20 D1.3 B cells that were transfected with control siRNA or treated with CK-689, the normalized average distance from the center decreased over time, indicating that microclusters moved towards the center of the immune synapse. This can be seen in kymographs depicting microclusters moving centripetally from the cell periphery and merging with other microclusters at the center of the immune synapse in *Figure 2—figure supplement 4G,H*. However, in Arp3 siRNA-transfected and CK-666-treated A20 D1.3 B cells, the average distance of BCR microclusters from the center of the synapse remained relatively unchanged over the entire 14 min imaging period. This is reflected in the kymographs in *Figure 2—figure supplement 4G,H*. To assess cSMAC formation, the distribution of Ag fluorescence into clusters

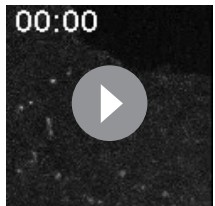

**Video 1.** BCR-Ag microclusters coalesce into a cSMAC at the B cell-APC contact site. A20 D1.3 B cells were added to COS-7 APCs expressing mHEL-GFP (white) and the B cell-APC contact site was imaged using spinning disk microscopy. Images taken every 12 s. Video playback is 10 frames per second (120X real speed). See also *Figure 1B*.

DOI: https://doi.org/10.7554/eLife.44574.004

over time was quantified. Cumulative frequency distribution curves showed that the 50th percentile (cumulative frequency = 0.5) for the number of clusters per cell required to contain 90% of the Ag fluorescence after 10 min of B cell-APC contact was ~1 for the control siRNA-transfected A20 D1.3 B cells and CK-689-treated cells versus four for the Arp3 siRNA-transfected cells and CK-666-treated cells (*Figure 2—figure supplement 4I,J*). For example, after 10 min of contact with an APC, the control siRNA-transfected B cell shown in *Figure 2A* had coalesced all of the Ag into four clusters, one of which contained 97.7% of the Ag fluorescence (*Figure 2C*). By contrast, a similar amount of Ag was distributed among 16 distinct clusters in the Arp3 siRNA-transfected cell depicted in *Figure 2A*, and no individual cluster contained more than 43% of the total Ag fluorescence (*Figure 2C*). Similar results were obtained when comparing the CK-689-treated (control) and CK-666-treated cells from *Figure 2B* (*Figure 2D*). Again, the control cell formed a cSMAC into which ~ 95% of the Ag was gathered whereas no cluster in the CK-666-treated cells contained more than 22% of the Ag fluorescence. For all B cells analyzed in this manner, the number of BCR-Ag microclusters formed after 10 min in contact with an APC, as well as the distribution of the total Ag fluorescence among individual clusters, is shown in *Figure 2—figure supplement 5*.

The failure to form a cSMAC when the Arp2/3 complex is depleted or inhibited was due to decreased merger of individual BCR microclusters into larger clusters. Consistent with this idea, the average size of BCR-Ag microclusters increased steadily over time in control cells as the microclusters merged but remained relatively unchanged over the 14 min of imaging in the Arp3 siRNA-treated cells and CK-666-treated cells (*Figure 2E,F*). Consequently, because microclusters merged, the number of individual microclusters per cell decreased over time in the control cells but remained similar, or increased, when Arp2/3 complex activity was inhibited (*Figure 2—figure supplement 4C, D*). Thus, BCR-mediated gathering of APC-bound Ags into a cSMAC at the center of the synapse is strongly dependent on the functions of the Arp2/3 complex.

## The Arp2/3 complex is important for BCR-induced actin reorganization and dynamics

To understand how the Arp2/3 complex promotes BCR microcluster centralization and cSMAC formation, we sought to define its role in actin organization and dynamics at the contact site between a

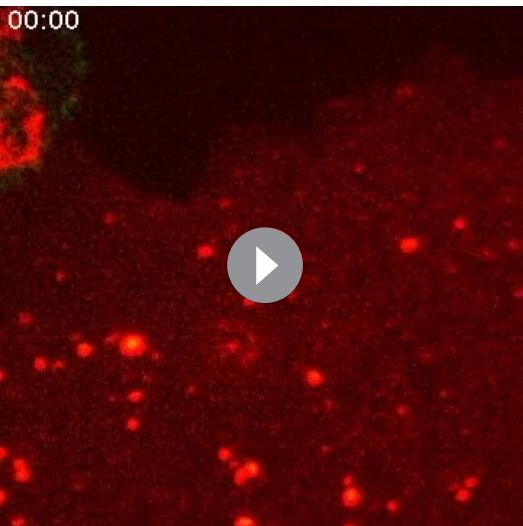

**Video 2.** Dynamic actin-based protrusions and BCR-Ag microclusters at the B cell-APC contact site. A20 D1.3 B cells expressing F-Tractin-GFP (green) were added to COS-7 APCs expressing mHEL-HaloTag (red) and the B cell-APC contact site was imaged using ISIM. Images taken every 6.6 s. Video playback is 10 frames per second (66X real speed). See also *Figure 1C*.
DOI: https://doi.org/10.7554/eLife.44574.005

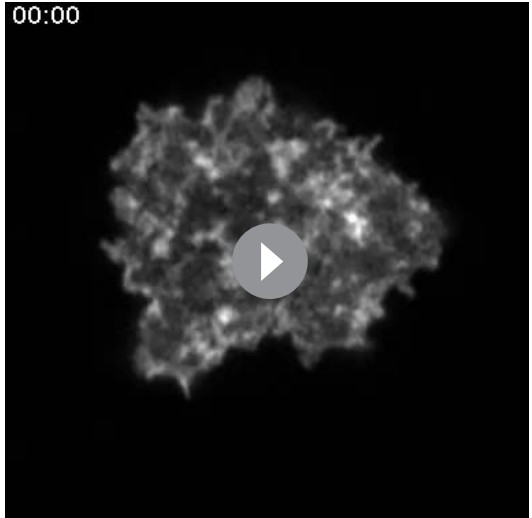

**Video 3.** Actin dynamics in B cells spreading on immobilized anti-IgG antibodies. A20 B cells expressing F-Tractin-GFP were added to anti-IgG-coated coverslips and the contact site was imaged using a Zeiss Airyscan microscope. Images taken every 2 s. Video playback is 15 frames per second (30X real speed). See also *Figure 1F*.
DOI: https://doi.org/10.7554/eLife.44574.006

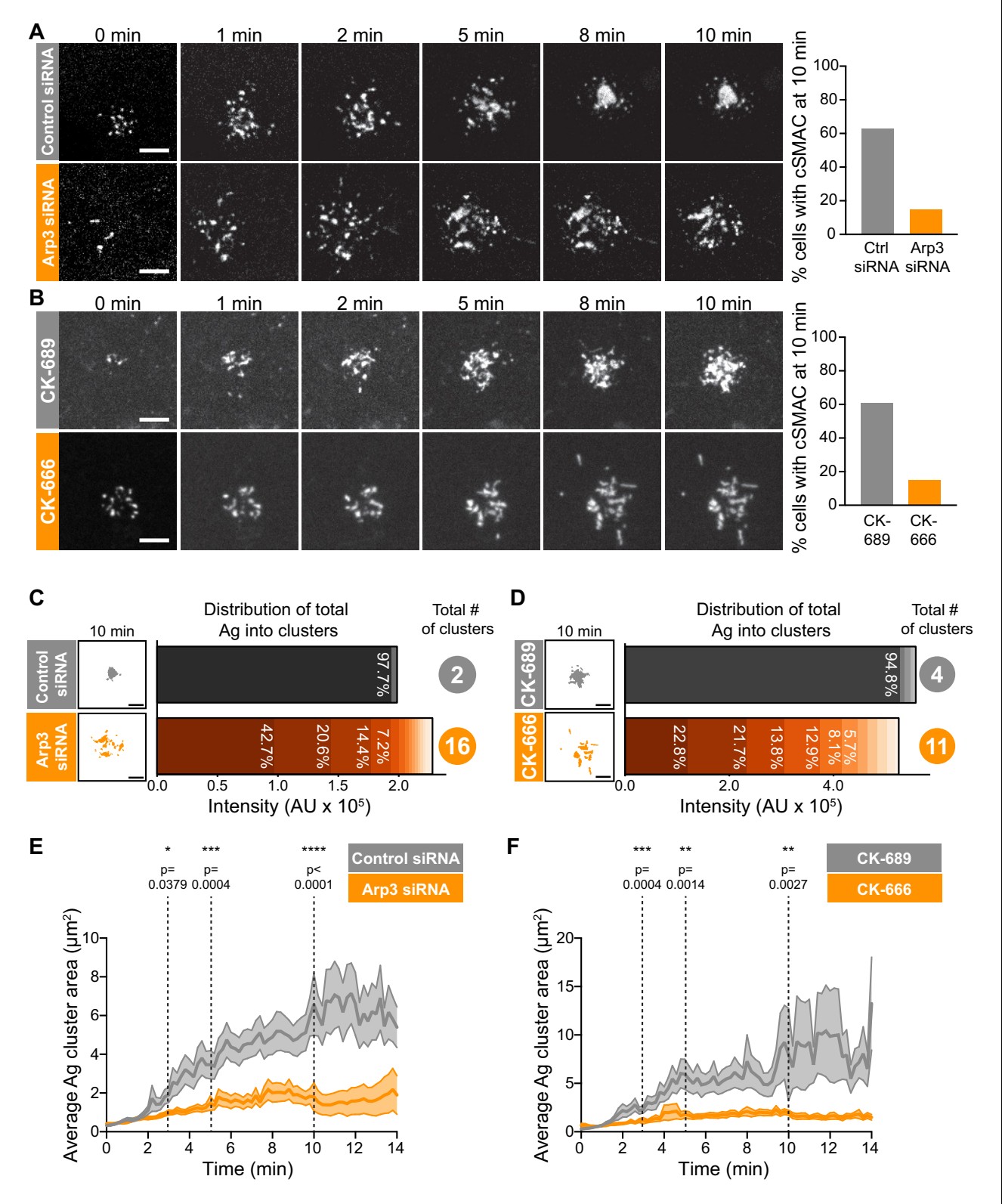

**Figure 2.** Arp2/3 complex function is important for centralization of BCR-Ag microclusters. (**A,B**) A20 D1.3 B cells were transfected with control siRNA or Arp3 siRNA (**A**) or pre-treated for 1 hr with 100 µM of either the control compound CK-689 or the Arp2/3 complex inhibitor CK-666 (**B**). The cells were then added to mHEL-GFP-expressing COS-7 APCs and the B cell-APC contact site was imaged every 12 s for 10 min by spinning disk microscopy. Images are from *Video 4* (Control siRNA) and *Video 5* (Arp3 siRNA) or *Video 6* (CK-689) and *Video 7* (CK-666). The percent of cells for which >90% of

*Figure 2 continued*

the total Ag fluorescence intensity was contained in one or two clusters is graphed. n > 35 cells from four independent experiments combined (**A**); n > 18 cells from three independent experiments combined (**B**). (**C,D**) For the control siRNA- and Arp3 siRNA-transfected cells in (**A**), or the CK-689- and CK-666-treated cells in (**B**), binary representations of the Ag clusters present after 10 min of APC contact are shown. The graphs depict the percent of the total Ag fluorescence intensity, in arbitrary units (AU), present in individual clusters. The numbers to the right are the total number of Ag clusters at 10 min. (**E,F**) Control siRNA- and Arp3 siRNA-transfected A20 D1.3 B cells (**E**) or CK-689- and CK-666-treated A20 D1.3 B cells (**F**), were added to mHEL-GFP-expressing COS-7 APCs and imaged every 12 s for 14 min by spinning disk microscopy. For each B cell, the average size of the BCR-Ag microclusters was determined for every frame in the video. The data are plotted as the mean (line) ± SEM (shaded area). In (**E**), n = 35 cells (control) or 48 cells (Arp3 siRNA) from four experiments. In (**F**), n = 18 cells (CK-689) or 20 cells (CK-666) from three experiments. The Mann-Whitney U test was used to calculate p values. ****$p<0.0001$; ***$p<0.001$; **$p<0.01$; *$p<0.05$. Scale bars: 5 μm.
DOI: https://doi.org/10.7554/eLife.44574.007

The following figure supplements are available for figure 2:

**Figure supplement 1.** siRNA-mediated knockdown of Arp3 in A20 B cells.
DOI: https://doi.org/10.7554/eLife.44574.008

**Figure supplement 2.** CK-666 treatment does not decrease cell viability.
DOI: https://doi.org/10.7554/eLife.44574.009

**Figure supplement 3.** CK-666 does not alter actin organization or dynamics in the COS-7 APCs.
DOI: https://doi.org/10.7554/eLife.44574.010

**Figure supplement 4.** The Arp2/3 complex is important for the coalescence of BCR-Ag microclusters into a cSMAC.
DOI: https://doi.org/10.7554/eLife.44574.011

**Figure supplement 5.** Distribution of total Ag fluorescence intensity in individual clusters.
DOI: https://doi.org/10.7554/eLife.44574.012

B cell and an Ag-bearing surface. Ags or anti-Ig antibodies that are immobilized on coverslips are widely used model systems for imaging BCR-induced spreading and actin reorganization with high resolution. To assess the role of the Arp 2/3 complex in this response, we used stimulated emission depletion (STED) microscopy. A20 B cells that bound to anti-IgG coated coverslips extended broad lamellipodia and exhibited radial spreading (control siRNA cells in *Figure 3A*, *Figure 3—figure supplement 1*, CK-689-treated cells in *Figure 3B*). This cell morphology was characterized by a dense peripheral ring of highly branched F-actin surrounding an actin-depleted central area that contained thin actin filaments (*Figure 3A,B*, *Figure 3—figure supplement 1*). Within the peripheral branched actin network of the lamellipodia, linear actin filaments, which may be nucleated by formins, extended perpendicularly from the edge of the cell to the inner face of the peripheral actin ring (*Figure 3A,B*, *Figure 3—figure supplement 1A* [yellow arrowheads]). We also observed long, linear actin filaments that formed actin arcs running parallel to the inner face of the peripheral ring of branched actin (*Figure 3B* [green arrowheads]; *Figure 3E*, *Figure 3—figure supplement 1*), which were associated with myosin IIA (*Figure 3—figure supplement 1B*). These are similar to the actomyosin arcs that have been described in T cells (*Murugesan et al., 2016*).

This actin organization at the Ag contact site was strongly dependent on Arp2/3 complex activity. When we depleted Arp3 with siRNA (*Figure 3A*), or inhibited the Arp2/3 complex in A20 B cells using CK-666 (*Figure 3B*), BCR-induced spreading and actin reorganization were dramatically altered. Unlike the control A20 B cells, which formed a highly dense, sheet-like actin network at the periphery, Arp3-depleted and CK-666-treated B cells extended long and loosely packed filopodia-like fibers (*Figure 3A,B*). This aberrant actin reorganization resulted in substantially reduced cell spreading on anti-IgG-coated coverslips (*Figure 3C,D*).

Because branched actin structures that are nucleated by the Arp2/3 complex exert forces on the plasma membrane, we hypothesized that retrograde flow of actin would be disrupted in B cells treated with the Arp2/3 inhibitor. To test this, A20 B cells expressing F-tractin-GFP were treated with CK-689 or CK-666 and then added to anti-IgG-coated coverslips. Total internal reflection fluorescence structured illumination microscopy (TIRF-SIM) was used to image actin dynamics with high spatial and temporal resolution. In control CK-689-treated cells, the F-actin networks were highly dynamic at the cell edge, with actin retrograde flow rates of ~2.5 μm/min (*Figure 3E*, *Video 8*). In contrast, in B cells treated with the Arp2/3 complex inhibitor, the peripheral actin structures were nearly static (*Figure 3*, *Video 9*). Thus, Arp2/3 complex function is essential for BCR-induced actin reorganization and for peripheral actin dynamics.

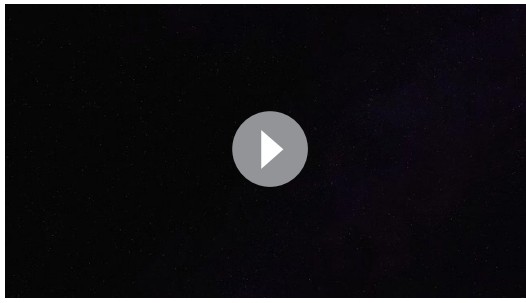 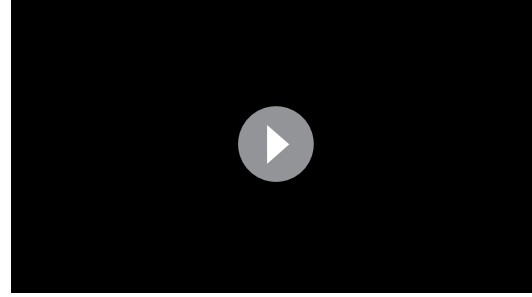

**Video 4.** BCR-Ag microclusters coalesce into a cSMAC in B cells expressing control siRNA. A20 D1.3 B cells expressing control siRNA were added to COS-7 APCs expressing mHEL-GFP (white) and the B cell-APC contact site was imaged using spinning disk microscopy. Images taken every 12 s. Video playback is 10 frames per second (120X real speed). Four representative cells are shown. The first cell in the video is also shown in *Figure 2A*. DOI: https://doi.org/10.5061/dryad.3b15k/2
DOI: https://doi.org/10.7554/eLife.44574.013

**Video 5.** Impaired cSMAC formation in B cells expressing Arp3 siRNA. A20 D1.3 B cells expressing Arp3 siRNA were added to COS-7 APCs expressing mHEL-GFP (white) and the B cell-APC contact site was imaged using spinning disk microscopy. Images taken every 12 s. The video is played back at 10 frames per second (120X real speed). Four representative cells are shown. The first cell in the video is also shown in *Figure 2A*. DOI: https://doi.org/10.5061/dryad.3bc215k/1
DOI: https://doi.org/10.7554/eLife.44574.014

## Arp2/3 complex-dependent actin dynamics and membrane retraction drive BCR microcluster centralization

How specific actin architectures allow B cells to scan APC membranes and then centralize the resulting BCR microclusters into a cSMAC is not known. To test the role of Arp2/3-nucleated actin networks in these processes, we used ISIM to simultaneously image actin structures and BCR-Ag microclusters in F-Tractin-GFP-transfected A20 D1.3 B cells interacting with mHEL-HaloTag-expressing COS-7 cells. As shown in *Video 2* and *Video 10*, control B cells extended dynamic, transient actin-rich protrusions over the surface of the APC, with both lamellipodial and filopodia-like structures being observed (see also *Figure 4A*, *Figure 5A*). Moreover, the high spatiotemporal resolution of ISIM revealed that BCR-Ag microclusters were embedded within actin-rich protrusions and moved

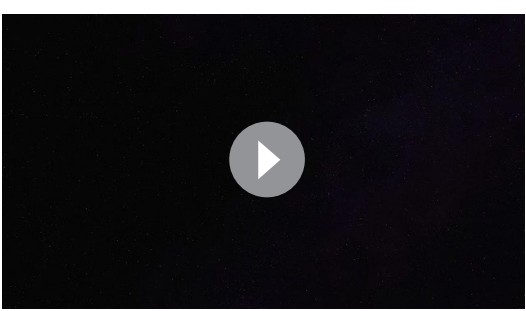 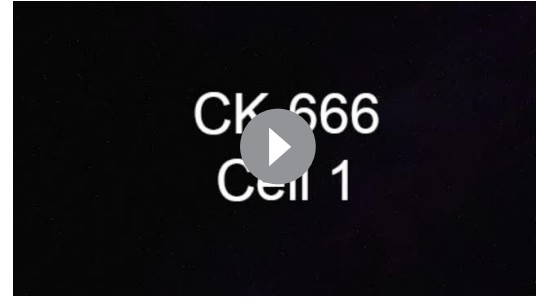

**Video 6.** BCR-Ag microclusters coalesce into a cSMAC in CK-689-treated B cells. A20 D1.3 B cells were pre-treated with 100 μM CK-689 for 1 hr and then added to COS-7 APCs expressing mHEL-GFP (white). The B cell-APC contact site was imaged using spinning disk microscopy. Images taken every 12 s. Video playback is 10 frames per second (120X real speed). Four representative cells are shown. The first cell in the video is also shown in *Figure 2B*. DOI: https://doi.org/10.5061/dryad.3bc215k/3
DOI: https://doi.org/10.7554/eLife.44574.015

**Video 7.** Impaired cSMAC formation in CK-666-treated B cells. A20 D1.3 B cells were pre-treated with 100 μM CK-666 for 1 hr and then added to COS-7 APCs expressing mHEL-GFP (white). The B cell-APC contact site was imaged using spinning disk microscopy. Images taken every 12 s. Video playback is 10 frames per second (120X real speed). Four representative cells are shown. The first cell in the video is also shown in *Figure 2B*. DOI: https://doi.org/10.5061/dryad.3bc215k/4
DOI: https://doi.org/10.7554/eLife.44574.016

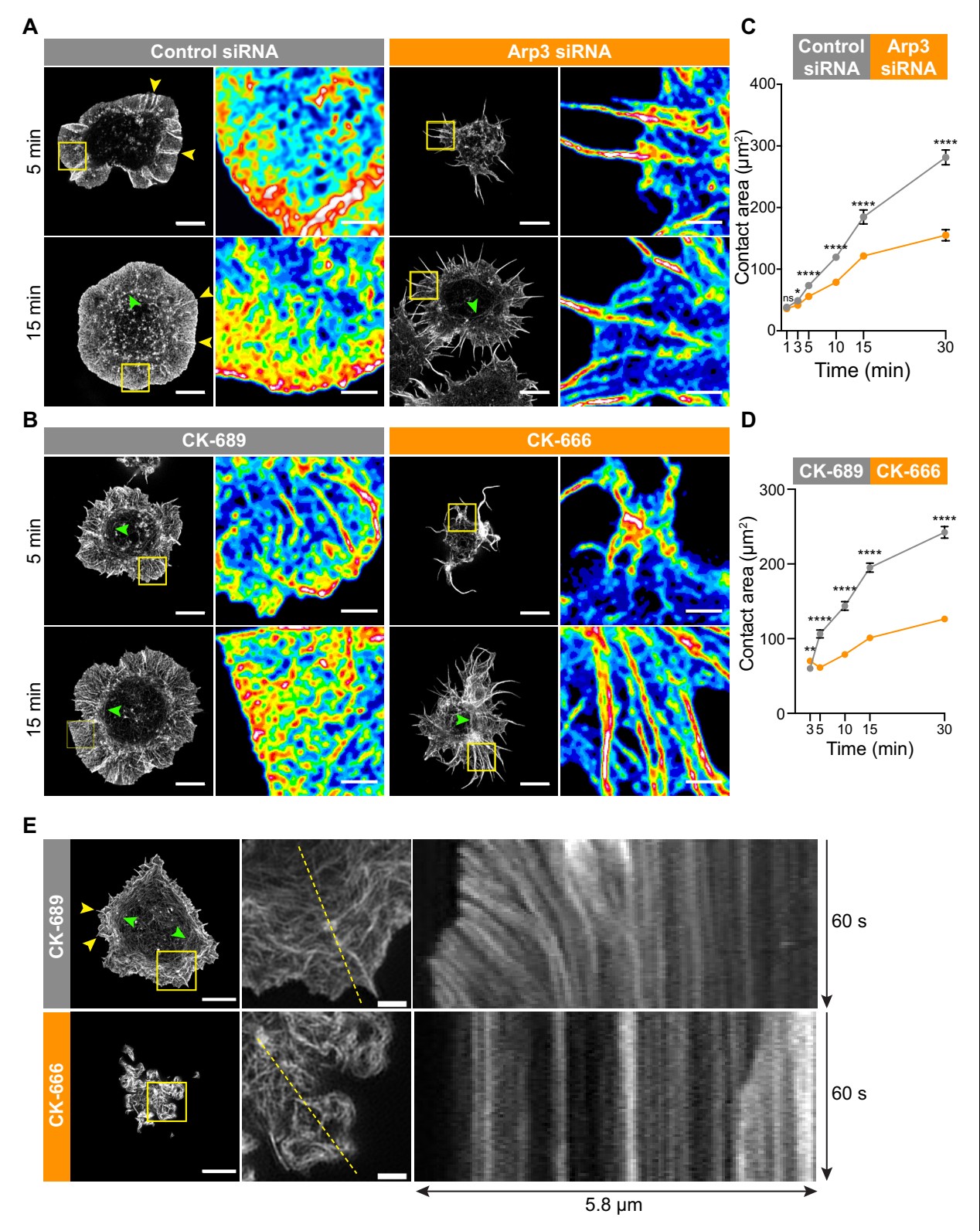

**Figure 3.** The Arp2/3 complex is important for BCR-induced actin reorganization and dynamics. (**A,B**) A20 B cells were transfected with either control siRNA or Arp3 siRNA (**A**) or pre-treated for 1 hr with 100 μM of the control compound CK-689 or the Arp2/3 complex inhibitor CK-666 (**B**). Cells were then allowed to spread on anti-IgG-coated coverslips for the indicated times before being fixed, stained for actin, and imaged by STED microscopy. Images representative of >20 cells per condition are shown. Yellow arrowheads indicate linear actin structures embedded within the peripheral

*Figure 3 continued on next page*

*Figure 3 continued*

branched actin network. Green arrowheads indicate actin arcs. Scale bars: 5 μm. For the regions within the yellow boxes, the images were enlarged (scale bars: 2 μm) and the relative densities of the actin structures are shown as heat maps. (C,D) A20 B cells were transfected with either control siRNA or Arp3 siRNA (C) or pre-treated for 1 hr with μM CK-689 or CK-666 (D). The cells were then allowed to spread on anti-IgG-coated coverslips for the indicated times before being stained for F-actin. The B cell-coverslip contact site was imaged using spinning disk microscopy. The cell area was quantified, using F-actin to define the cell edge. For each data point the mean ± SEM is shown for >27 cells from a representative experiment. ****p<0.0001; **p<0.01; *p<0.05; ns, not significant; Mann-Whitney U test. (E) A20 B cells expressing F-Tractin-GFP were pre-treated for 1 hr with CK-689 or CK-666 and then added to anti-IgG-coated chamber wells. After 5 min, the cells were imaged by TIRF-SIM at 1 s intervals for 1 min. The left panels are the first images from *Video 8* (CK-689-treated cells) and *Video 9* (CK-689-treated cells), respectively (Scale bars: 5 μm). Yellow arrowheads indicate linear actin structures embedded within the peripheral branched actin network. Green arrowheads indicate actin arcs. The middle panels are enlargements of the areas within the yellow boxes in the left panels (Scale bars: 2 μm). The right panels are kymographs along the yellow dotted lines in the middle panels.

DOI: https://doi.org/10.7554/eLife.44574.017

The following figure supplement is available for figure 3:

**Figure supplement 1.** Actin and myosin structures in A20 B cells spreading on immobilized anti-IgG.

DOI: https://doi.org/10.7554/eLife.44574.018

---

toward the cell body as these protrusions were retracted (*Figure 4A*, *Figure 5A*, *Figure 4—figure supplement 1*, *Video 10*; see also *Figure 1C*). For example, the microcluster highlighted in *Figure 5A* is encaged within an actin-rich protrusion. In contrast, B cells that were treated with the Arp2/3 complex inhibitor CK-666 primarily extended long, linear, filopodia-like actin structures over the surface of the APC (*Figure 4A*, *Figure 5A*, *Figure 4—figure supplement 1*, *Video 11*). Most of these actin structures did not retract toward the cell body over the 6.5 min observation period. Moreover, BCR-Ag microclusters that formed on these protrusions did not translocate toward the center of the synapse or merge with other microclusters (*Figure 4A*, *Figure 5A*, *Figure 4—figure supplement 1A*, *Video 11*).

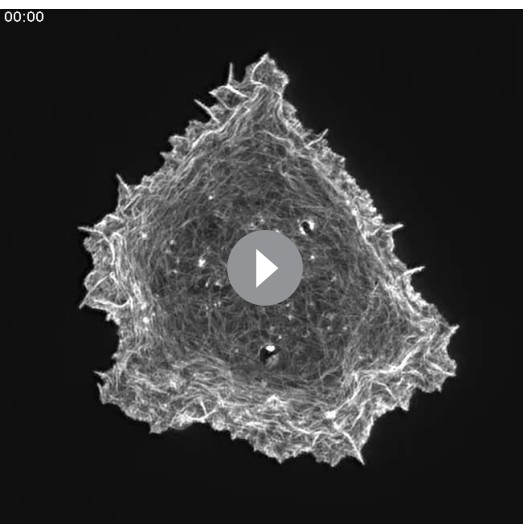

**Video 8.** Peripheral actin dynamics in CK-689-treated B cells plated on immobilized anti-IgG. A20 B cells expressing F-Tractin-GFP were pre-treated for 1 hr with 100 μM CK-689 and then added to anti-IgG-coated coverslips. The contact site was imaged using TIRF-SIM. Images taken every 1 s. Video playback is 10 frames per second (10X real speed). See also *Figure 3E*.

DOI: https://doi.org/10.7554/eLife.44574.019

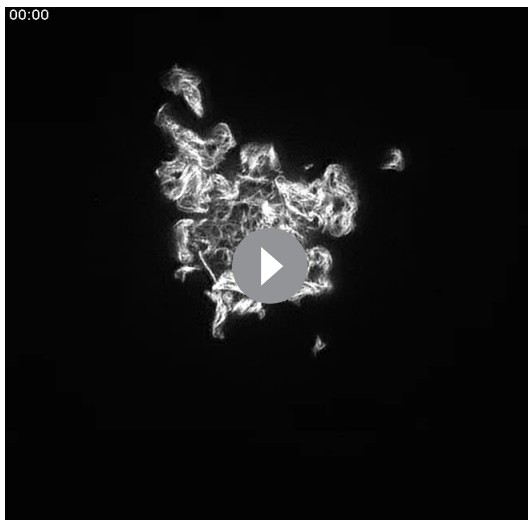

**Video 9.** Impaired peripheral actin dynamics in CK-666-treated B cells plated on immobilized anti-IgG. A20 B cells expressing F-Tractin-GFP were pre-treated for 1 hr with 100 μM CK-666 and then added to anti-IgG-coated coverslips. The contact site was imaged using TIRF-SIM. Images taken every 1 s. Video playback is 10 frames per second (10X real speed). See also *Figure 3E*.

DOI: https://doi.org/10.7554/eLife.44574.020

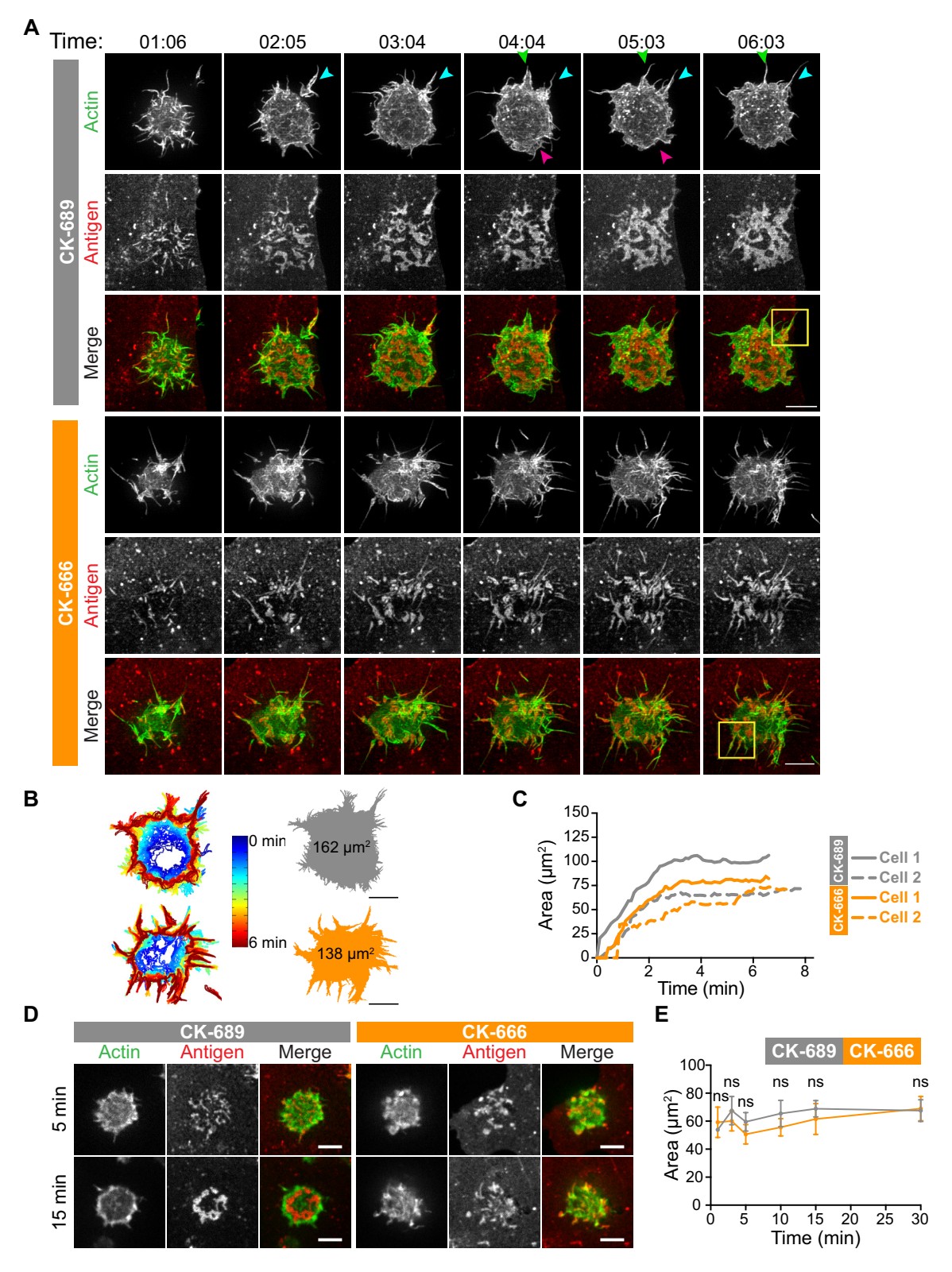

**Figure 4.** The Arp2/3 complex is important for actin and BCR microcluster dynamics at the B cell-APC immune synapse. (**A–C**) A20 D1.3 B cells expressing F-Tractin-GFP (green) were pre-treated with 100 μM CK-689 or CK-666 for 1 hr and then added to COS-7 cells expressing mHEL-HaloTag (red). The cells were imaged every 6.6 s for 6 min using ISIM. Images from **Video 10** and **Video 11** are shown in (**A**). Arrowheads indicate new protrusion events. The yellow boxes indicate the regions that are enlarged in **Figure 5**. In (**B**) the cell edge from each frame in **Video 10** (CK-689-

*Figure 4 continued on next page*

*Figure 4 continued*

treated cells) or *Video 11* (CK-666-treated cells), as defined by the peripheral F-actin, was overlaid as a temporally coded time series (left). The total area searched by the B cell over the 6-min period of observation is shown the right. In (C) the B cell-APC contact area is shown as a function of time for the cells in (A) (solid lines), and for another set of representative cells (dashed lines; images are shown in *Figure 4—figure supplement 1*). (D,E) A20 D1.3 B cells expressing F-Tractin-GFP (green) were pre-treated with 100 µM CK-689 or CK-666 and then added to COS-7 cells expressing mHEL-HaloTag (red). The cells were fixed at the indicated times and imaged by spinning disk microscopy. Representative images are shown (D) and the B cell-APC contact area (mean ± SEM), as defined by F-actin, is graphed (E). For each data point, n > 20 cells from three independent experiments except for the 1 min time point (n = 4 cells). ns, not significant (p=0.7219, p=0.1310, p=0.0722, p=0.1443, p=0.2930, p=0.6263 for the 1, 3, 5, 10, 15, and 30 min time points, respectively); two-tailed paired t-test. Scale bars: 5 µm.
DOI: https://doi.org/10.7554/eLife.44574.021

The following figure supplement is available for figure 4:

**Figure supplement 1.** Arp2/3 complex activity is required for actin dynamics and for the centripetal movement of BCR-Ag microclusters at the periphery of the B cell-APC contact site.
DOI: https://doi.org/10.7554/eLife.44574.022

Although the architectures of the actin protrusions generated by control and CK-666-treated cells were dramatically different, the cells explored roughly the same area of the APC surface over the period of observation (*Figure 4B–E*, *Figure 4—figure supplement 1B*). This is in contrast to the significantly reduced cell spreading on rigid anti-Ig-coated surfaces that was observed when the Arp2/3 complex was depleted or inhibited (*Figure 3*). Thus, Arp2/3 complex activity is requred for B cells to spread on rigid surfaces but may be dispensable for B cells to effectively probe the APC surface and form BCR-Ag microclusters under the conditions tested. However, Arp2/3 complex-mediated actin polymerization, and the resulting actin retrograde flow, appears to be essential for the centralization of BCR-Ag microclusters. Consistent with this idea, kymograph analysis showed that peripheral BCR-Ag clusters in control cells moved toward the center of the synapse together with, and at a similar velocity as, the surrounding actin structures (*Figure 5B*, *Figure 4—figure supplement 1C*). In contrast, when the Arp2/3 complex was inhibited, both the BCR-Ag clusters and the associated actin structures were relatively immobile (*Figure 5B*, *Figure 4—figure supplement 1C*). Taken together, these data suggest that Arp2/3 complex-dependent actin structures encage BCR-Ag microclusters, such that the actin retrograde flow drives their initial centripetal movement. This appears to be required for the subsequent formation of a cSMAC, which occurs in the actin-depleted central region of the cell.

## Arp2/3 complex activity amplifies BCR signaling at the immune synapse

How the spatial organization of the BCR at the immune synapse impacts BCR signaling and B cell activation is not fully understood. Because inhibition of the Arp2/3 complex prevented the centripetal movement of BCR microclusters into a cSMAC but did not significantly alter the amount of Ag gathered into BCR microclusters (*Figure 2C*, *Figure 2—figure supplement 4A,B*, *Figure 6—figure supplement 1A*), this allowed us to ask whether the Arp2/3 complex-dependent spatial patterning of BCR-Ag microclusters impacts BCR signaling output. We quantified proximal BCR signaling events at the contact site between ex vivo primary murine splenic B cells and COS-7 APCs expressing a single-chain anti-Igκ surrogate Ag on their surface (*Freeman et al., 2011*). A critical initial step in BCR signaling is phosphorylation of the tyrosine residues within the immunoreceptor tyrosine-based activation motifs (ITAMs) present in the CD79a/b signaling subunit of the BCR (*Dal Porto et al., 2004*). This is required for the recruitment and activation of Syk, a tyrosine kinase that phosphorylates multiple proteins that are critical for BCR signaling and B cell activation.

APC-induced phosphorylation of CD79a/b at the immune synapse was assessed by staining with an antibody that recognizes the phosphorylated ITAMs of both CD79a and CD79b. We found that CD79 phosphorylation occurred rapidly at the B cell-APC contact site and co-localized with BCR-Ag clusters, which were detected using an antibody that detects the surrogate Ag (*Figure 6A*). As shown in *Figure 2*, the BCR-Ag microcluster coalesced into a tight cSMAC within 5–10 min in control cells but not in B cells treated with the Arp2/3 complex inhibitor CK-666 (*Figure 6A*). Using quantitative image analysis, we then determined the relationship between the amount of Ag gathered into clusters and the signaling output at those BCR-Ag microclusters. For each B cell, the total phospho-CD79 (pCD79) or phospho-Syk (pSyk) fluorescence intensity present in clusters at the B cell-APC

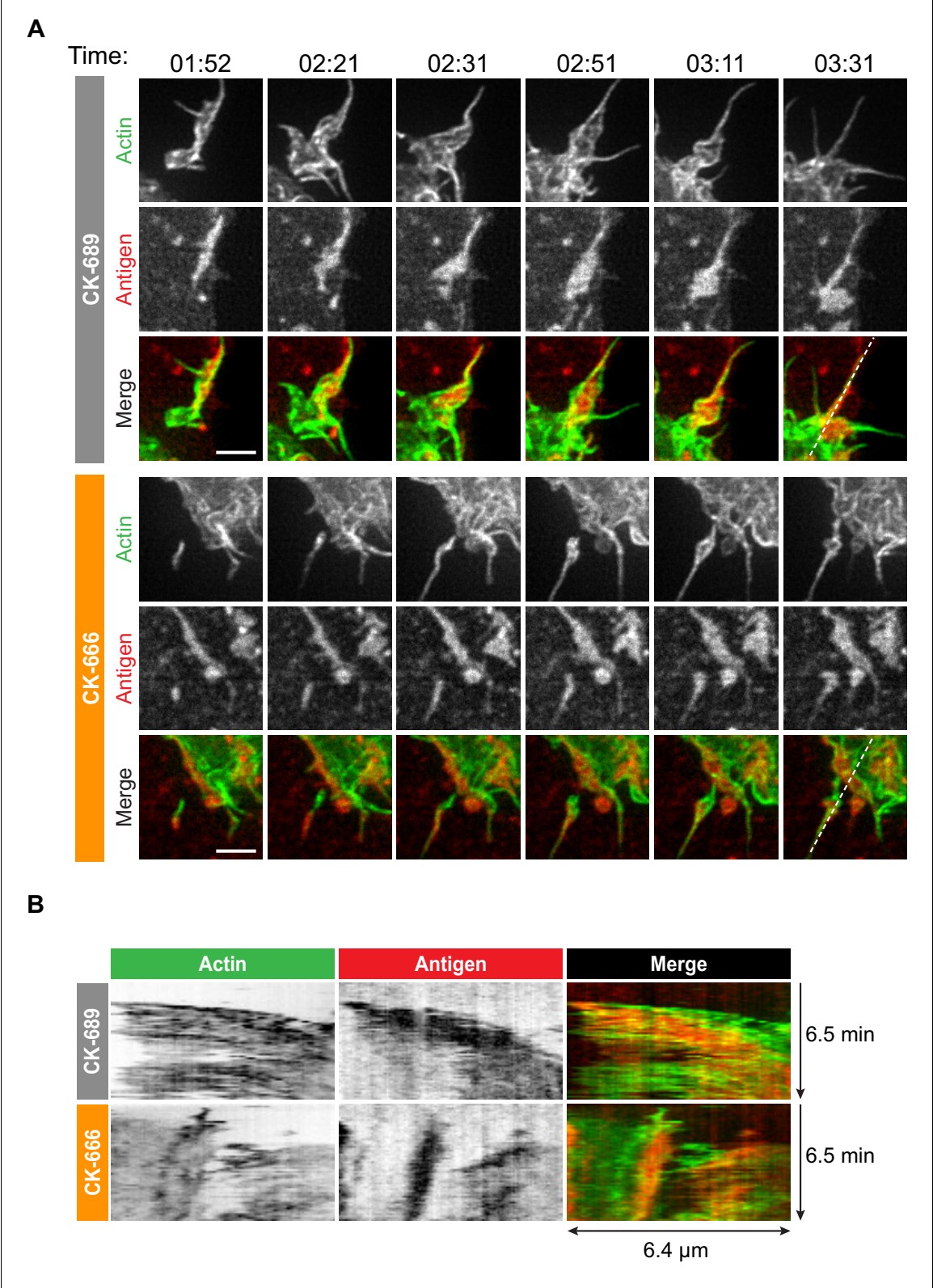

**Figure 5.** Arp2/3 complex-dependent actin structures encage BCR-Ag microclusters on membrane protrusions. (**A**) Images from *Video 10* and *Video 11* showing enlargements of the regions indicated by the yellow boxes in *Figure 4*. A20 D1.3 B cells expressing F-Tractin-GFP (green) were pre-treated with 100 μM CK-689 or CK-666 for 1 hr and then added to COS-7 cells expressing mHEL-HaloTag (red). The cells were imaged every 6.6 s for 6 min using ISIM. Scale bars: 2 μm. (**B**) The kymographs represent a time series of images taken along the white dashed lines in (**A**).
DOI: https://doi.org/10.7554/eLife.44574.023

interface was divided by the total fluorescence intensity of clustered Ag. In control B cells treated with CK-689, pCD79 levels were maximal at 5 and 10 min after the B cells were added to the APCs and declined thereafter (*Figure 6B*), perhaps due to the internalization of BCR-Ag microclusters. Importantly, B cells treated with the Arp2/3 complex inhibitor exhibited significantly lower pCD79 levels at the 5, 10, and 15 min time points compared to control cells (*Figure 6B*, *Figure 6—figure supplement 1B*). Similar results were obtained when HEL-specific B cells from MD4 mice were added to COS-7 APCs expressing the HEL-HaloTag Ag (*Figure 6—figure supplement 1C,D*).

We also analyzed CD79 phosphorylation by immunoblotting, which would detect BCRs on the cell surface as well as ones that have been internalized. Indeed, immunoblotting showed that APC-induced CD79 phosphorylation increased continually over the first 30 min of B cell-APC contact (*Figure 6C*). Importantly, similar to what was observed by imaging the B cell-APC interface, treating the B cells with CK-666 reduced APC-induced CD79 phosphorylation, especially at the 15 and 30 min time points (*Figure 6C*, *Figure 6—figure supplement 2A*). In contrast, CK-666 treatment did not impair the ability of soluble anti-Ig antibodies to stimulate CD79 phosphorylation (*Figure 6D*, *Figure 6—figure supplement 2B*). This suggests that the Arp2/3 complex-dependent spatial patterning of BCR-Ag microclusters amplifies proximal BCR signaling in response to polarized, membrane-bound Ags but not soluble Ags.

The Ag-induced binding of Syk to phosphorylated CD79a/b leads to the phosphorylation of Syk (*Rowley et al., 1995*). Phosphorylation on Y342 and Y346 increases Syk activity and generates binding sites for PLCγ2 and Vav (*Geahlen, 2009*). When ex vivo primary B cells interacted with APCs expressing the single-chain anti-Igκ surrogate Ag, co-localization of pSyk (Y346) with BCR-Ag clusters was observed within 3 min (*Figure 6E*, *Figure 6—figure supplement 3A*). B cells treated with CK-666 had significantly lower levels of pSyk at 3 min and 5 min after contacting the APCs, compared to control cells (*Figure 6F*). Similar results were obtained when HEL-specific primary B cells were added to APCs expressing the HEL-HaloTag Ag (*Figure 6—figure supplement 3B,C*). Together, these data suggest that the Arp2/3 complex is required for the amplification of the

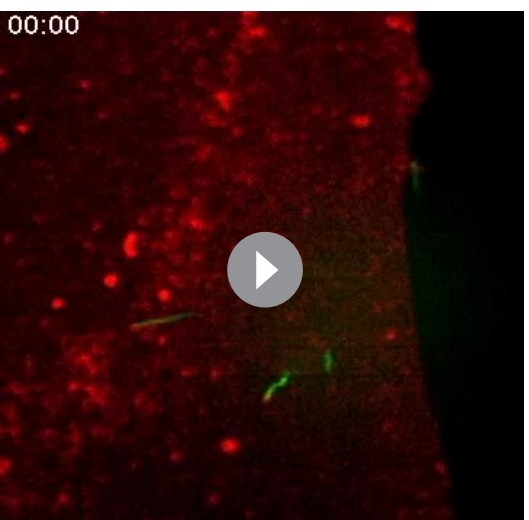

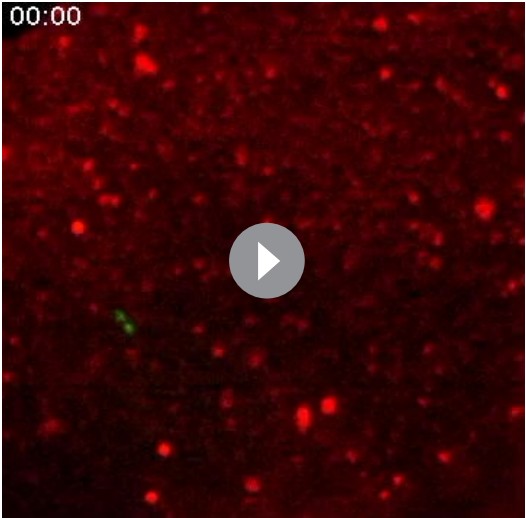

**Video 10.** Actin and BCR-Ag microcluster dynamics in CK-689-treated B cells interacting with APCs. A20 D1.3 B cells expressing F-Tractin-GFP (green) were pre-treated for 1 hr with 100 µM CK-689 and then added to COS-7 APCs expressing mHEL-HaloTag (red). The B cell-APC contact site was imaged using ISIM. Images taken every 6.6 s. Video playback is 10 frames per second (66X real speed). See also *Figure 4* and *Figure 5*.
DOI: https://doi.org/10.7554/eLife.44574.024

**Video 11.** Impaired actin and BCR-Ag microcluster dynamics in CK-666-treated B cells interacting with APCs. A20 D1.3 B cells expressing F-Tractin-GFP (green) were pre-treated for 1 hr with 100 µM CK-666 and then added to COS-7 APCs expressing mHEL-HaloTag (red). The B cell-APC contact site was imaged using ISIM. Images taken every 6.6 s. Video playback is 10 frames per second (66X real speed). See also *Figure 4* and *Figure 5*.
DOI: https://doi.org/10.7554/eLife.44574.025

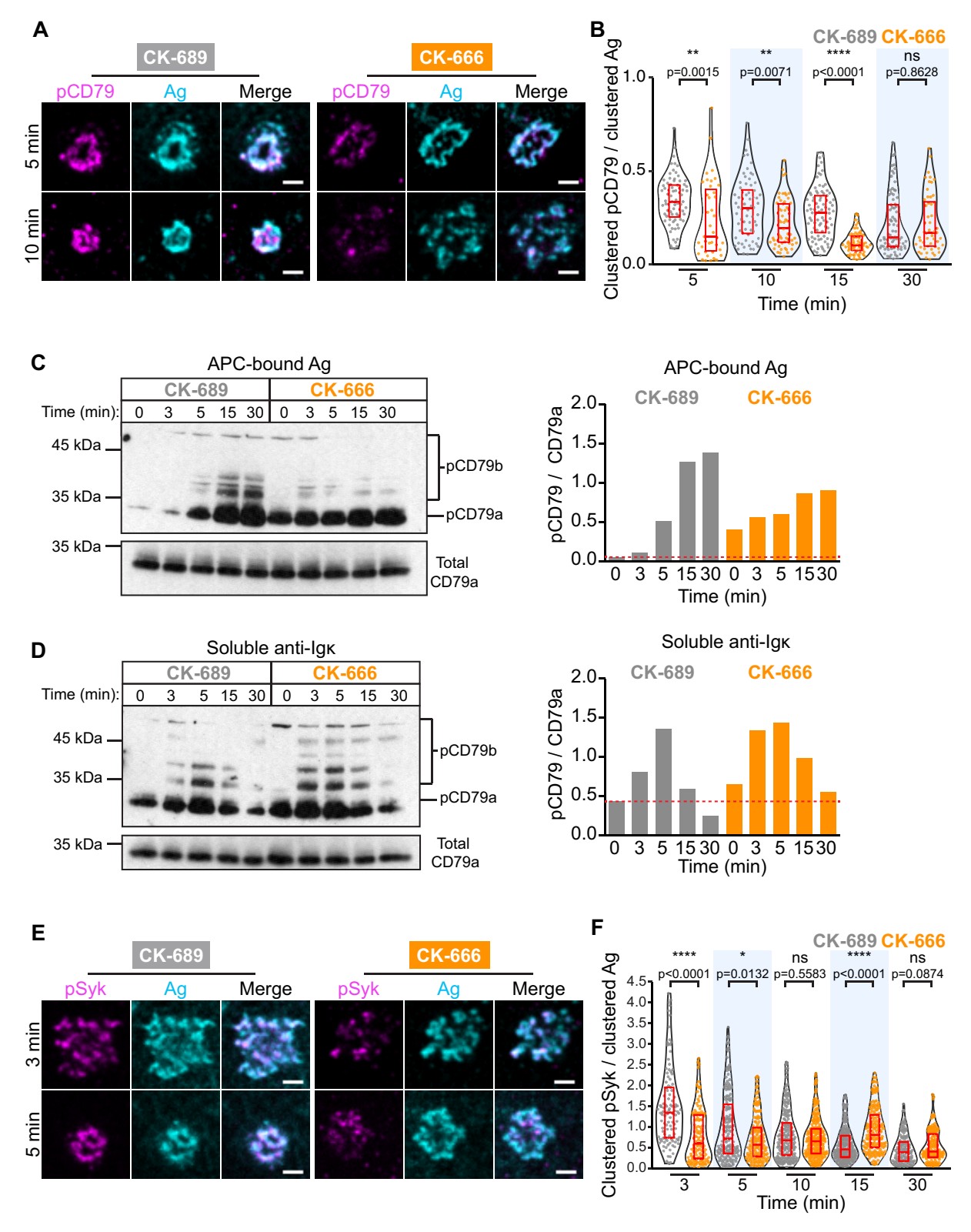

**Figure 6.** The Arp2/3 complex amplifies proximal BCR signaling. (**A,B**) Primary murine B cells were pre-treated with 100 µM CK-689 or CK-666 for 1 hr and then added to COS-7 cells expressing the single-chain anti-Igκ surrogate Ag. The cells were fixed at the indicated times and stained with an antibody that recognizes the surrogate Ag and with an antibody that recognizes the phosphorylated ITAMs in CD79a and CD79b (pCD79). Images of representative cells are shown (**A**). For each B cell, the total fluorescence intensity of clustered pCD79 was divided by the total fluorescence intensity of

*Figure 6 continued on next page*

*Figure 6 continued*
clustered Ag at the B cell-APC contact site. Beeswarm plots in which each dot is one cell. The median (red line) and interquartile ranges (red box) for >39 cells for each time point from a representative experiment are shown (B). (C,D) Primary murine B cells were pre-treated with 100 µM CK-689 or CK-666 for 1 hr and then added to COS-7 cells expressing the single-chain anti-Igκ surrogate Ag (C) or stimulated with 10 µg/ml soluble anti-Igκ (D) for the indicated times. pCD79 and total CD79a immunoblots are shown (left panels) and the pCD79/total CD79a ratios are graphed (right panels). Dotted red line corresponds to the pCD79/total CD79a ratio value for unstimulated CK-689-treated B cells. Representative data from one of seven experiments. An additional independent experiment is shown in *Figure 6—figure supplement 2*. See *Figure 9—figure supplement 6* for full blots. (E, F) Primary murine B cells that had been pre-treated with 100 µM CK-689 or CK-666 for 1 hr were added to COS-7 cells expressing the single-chain anti-Igκ Ag. The cells were fixed at the indicated times and stained for the surrogate Ag and pSyk (E). For each B cell, the total fluorescence intensity of clustered pSyk was divided by the total fluorescence intensity of clustered Ag at the B cell-APC contact site. Beeswarm plots with the median and interquartile ranges for >112 cells for each time point from a representative experiment are shown (F). ****p<0.0001; ***p<0.001; **p<0.01; *p<0.05; ns, not significant; Mann-Whitney U test. Scale bars: 2 µm.
DOI: https://doi.org/10.7554/eLife.44574.026
The following figure supplements are available for figure 6:

**Figure supplement 1.** Arp2/3 complex activity amplifies CD79 phosphorylation.
DOI: https://doi.org/10.7554/eLife.44574.027
**Figure supplement 2.** Arp2/3 complex activity amplifies Syk phosphorylation.
DOI: https://doi.org/10.7554/eLife.44574.028
**Figure supplement 3.** Additional independent experiment associated with *Figure 6C,D*.
DOI: https://doi.org/10.7554/eLife.44574.029

earliest BCR signaling events, phosphorylation of the CD79a/b ITAMs and phosphorylation of tyrosine residues in Syk that are required for its activation.

## Inhibiting the Arp2/3 complex increases tonic BCR signaling and BCR diffusion

Although treating B cells with the Arp2/3 complex inhibitor reduced the ability of APC-bound Ags to stimulate the phosphorylation of CD79 and Syk, CK-666-treated B cells exhibited higher levels of pCD79 prior to Ag encounter (*Figure 6C,D*; 0 min time point). This was accompanied by increased phosphorylation of ERK and Akt, downstream targets of BCR signaling (*Figure 7A*). Actin-based diffusion barriers restrict the lateral mobility of the BCR within the plasma membrane and increased BCR mobility is associated with increased tonic Ag-independent BCR signaling (*Freeman et al., 2015*; *Treanor et al., 2010*). Because Arp2/3 complex activity contributes to the formation of the submembrane cortical actin network, which creates diffusion barriers for transmembrane proteins, we tested the hypothesis that Arp2/3 complex activity limits Ag-independent BCR signaling by restricting BCR mobility prior to Ag encounter. We used single-particle tracking (SPT) to compare the diffusion and confinement properties of both IgM- and IgD-containing BCRs, in control versus CK-666-treated primary B cells. Indeed, the median diffusion coefficients for IgM and IgD were approximately 2.8-fold higher in CK-666-treated cells than in control CK-689-treated cells (*Figure 7B–D*). Consistent with their increased lateral mobility, both IgM and IgD had larger effective confinement diameters in CK-666-treated cells than in control CK-689-treated cells (*Figure 7B–D*). Thus, CK-666 treatment increases both BCR mobility and tonic BCR signaling. This suggests that the Arp2/3 complex contributes to the formation of actin-based barriers that normally limit BCR diffusion and tonic BCR signaling.

## Arp2/3 complex activity is important for BCR-CD19 interactions

CD19 is essential for B cell activation by membrane-bound Ags but not soluble Ags (*Depoil et al., 2008*; *Xu et al., 2014*). Initial BCR signaling leads to phosphorylation of CD19 on key tyrosine residues, allowing CD19 to recruit PI3K and Vav (*Buhl et al., 1997*; *O'Rourke et al., 1998*; *Tuveson et al., 1993*). CD19 is relatively immobile within the plasma membrane and this facilitates BCR-CD19 interactions and CD19-dependent amplification of BCR signaling (*Mattila et al., 2013*). Indeed, depolymerizing the actin cytoskeleton results in diminished CD19 phosphorylation and PI3K signaling in response to BCR stimulation (*Keppler et al., 2015*; *Mattila et al., 2013*). Using SPT, we found that treating B cells with CK-666 caused a 2.3-fold increase in the median diffusion coefficient for CD19 and increased its effective confinement diameter (*Figure 8A*). This suggests that Arp2/3

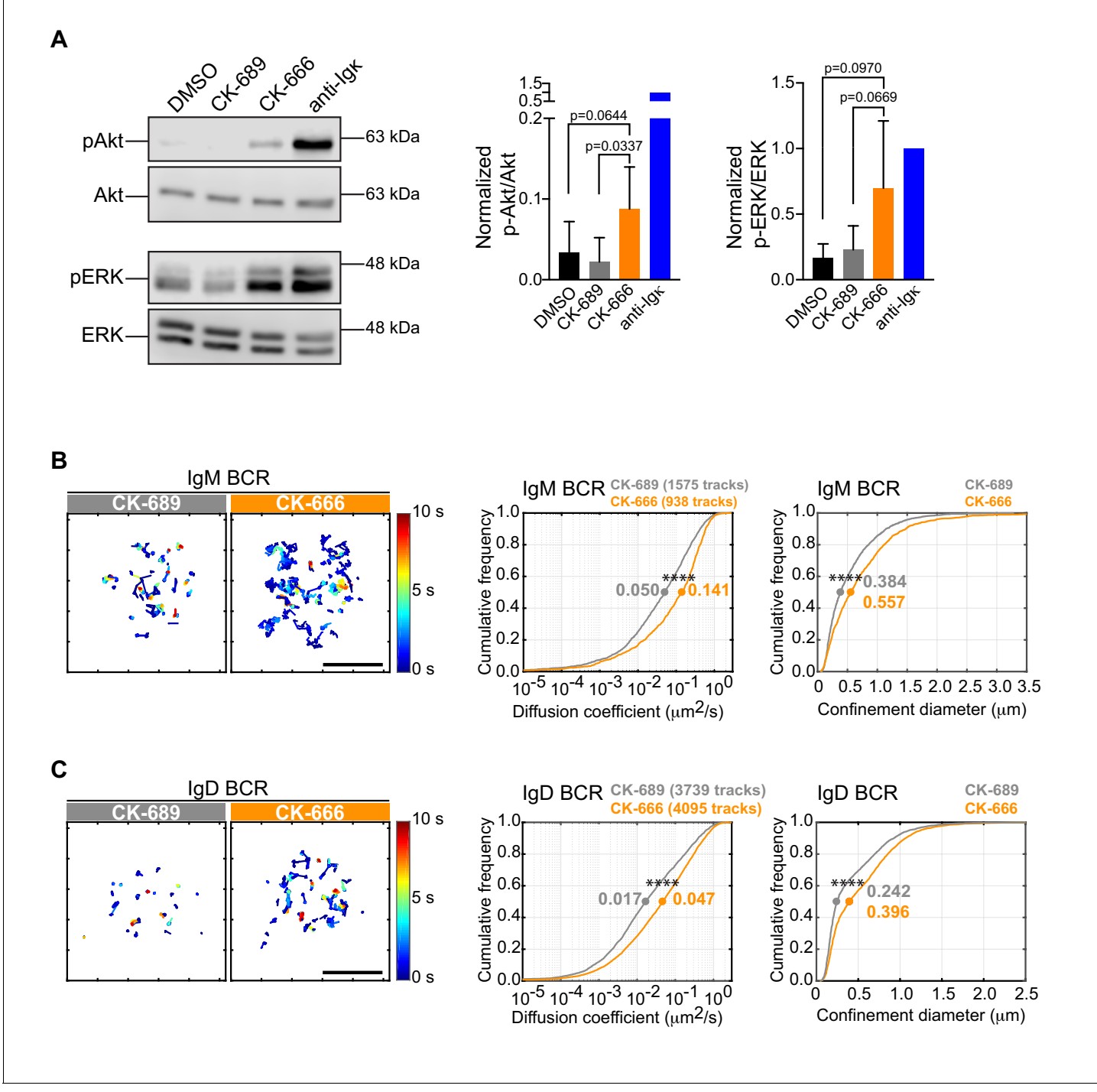

**Figure 7.** The Arp2/3 complex regulates BCR mobility and tonic signaling in resting B cells. (A) Ex vivo primary murine B cells were treated with DMSO, 100 µM CK-689, or 100 µM CK-666 for 1 hr, or stimulated with anti-Igκ antibodies for 5 min. Cell extracts were analyzed by immunoblotting with antibodies against phospho-ERK (pERK) and total ERK, or phospho-Akt (pAkt) and total Akt. Representative blot are shown (left). Band intensities were quantified and the ratio of pERK/total ERK and pAkt/total Akt (right) relative to those in anti-Igκ-treated cells (=1.0) are graphed as the mean ± SEM for four (pERK) or five (pAkt) independent experiments. Two-tailed paired t-test. See *Figure 9—figure supplement 6* for full blots. (B–D) Ex vivo primary murine splenic B cells were treated with 100 µM CK-689 or CK-666 for 1 hr. SPT was then carried out by labeling the cells at low stoichiometry with Cy3-labeled Fab fragments of antibodies to IgM (B) or IgD (C). The cells were then settled onto non-stimulatory anti-MHC II-coated coverslips and imaged for 10 s at 33 Hz by total internal reflection florescence microscopy (TIRFM). Single-particle trajectories from representative cells are plotted using a color-coded temporal scale (left panels). Scale bars: 5 µm. Diffusion coefficients were calculated for the indicated number of tracks and cumulative frequency curves are shown (center panels). The diameter of maximum displacement over the 10 s period of observation (confinement diameter) was

*Figure 7 continued on next page*

*Figure 7 continued*

calculated for each track and cumulative frequency curves are shown (right panels). The dots on the curves indicate the median values. Representative data from one of three independent experiments ****p<0.0001; Kolmogorov-Smirnov test.

DOI: https://doi.org/10.7554/eLife.44574.030

complex-dependent actin structures limit the lateral mobility of CD19 in the plasma membrane, which may be important for membrane-bound Ags to stimulate CD19 phosphorylation and augment BCR-CD19 interactions.

To test this, we asked whether inhibiting Arp2/3 complex activity altered the ability of membrane-bound Ags to stimulate CD19 phosphorylation. We used phospho-specific antibodies to detect CD19 that is phosphorylated on Y531 (pCD19) and quantified the level of pCD19 at the contact site between primary splenic B cells and APCs expressing the single-chain anti-Igκ surrogate Ag (*Figure 8B,C*). We found that the initial peak of APC-induced CD19 phosphorylation was diminished and delayed in CK-666-treated B cells, compared to the control CK-689-treated cells (*Figure 8C*, *Figure 8—figure supplement 1A*). As observed for phosphorylation of CD79, inhibiting the Arp2/3 complex did not impair the ability of soluble anti-Ig antibodies to stimulate rapid and robust CD19 phosphorylation (*Figure 8—figure supplement 1B*). This suggests that Arp2/3 complex-dependent actin structures provide spatial organization that is important for BCR-induced CD19 phosphorylation in response to spatially restricted membrane-bound Ags.

Because the co-localization of pCD19 with the BCR could determine the extent to which CD19 amplifies BCR signaling, we next quantified the fraction of total pCD19 fluorescence that overlaps BCR-Ag microclusters, using the Manders' coefficient. In control cells, much of the pCD19 fluorescence occured within BCR-Ag clusters at all time points after initiating B cell-APC contact (*Figure 8D*). The pCD19-Ag overlap was significantly less in the CK-666-treated B cells than in the control cells at all time points. By contrast, the overlap between pSyk clusters and BCR-Ag clusters in CK-666-treated cells was either not significantly different from, or was slightly higher, than in control cells over the first 15 min of B-APC interactions (*Figure 8—figure supplement 1C*). Thus, unlike pSyk, which is strongly associated with signaling BCRs, the co-localization of pCD19 with the BCR is impacted by Arp2/3 complex activity. Importantly, we found that the total amount of pCD19 fluorescence that overlapped with BCR-Ag clusters was much lower in CK-666-treated cells than in control cells at the 3-min time point. This reflects both decreased CD19 phosphorylation and decreased BCR-pCD19 overlap in the CK-666-treated cells. As for total pCD19 fluorescence, the initial peak of pCD19 fluorescence within BCR-Ag microclusters was diminished and delayed when Arp2/3 complex activity was inhibited (*Figure 8E*). Taken together, these findings indicate that actin networks nucleated by the Arp2/3 complex promote APC-induced CD19 phosphorylation and enhance the interaction of pCD19 with the BCR.

## Arp2/3 complex activity is important for BCR-induced B cell activation responses

We next sought to determine if the early BCR signaling defects observed in B cells treated with the Arp2/3 complex inhibitor translated into reduced B cell activation responses at later times. In *Figure 9*, we show that COS-7 APCs expressing the single chain anti-Igκ surrogate Ag induce primary B cells to undergo robust transcriptional responses, activation marker upregulation, cell size increase, and proliferation. We then assessed whether inhibiting Arp2/3 complex activity by treating the cells with CK-666 impacted these BCR-mediated responses that occur after 3–72 hr of Ag stimulation (*Figure 9A*).

To assess BCR-induced transcriptional responses, we used B cells from Nur77^GFP reporter mice in which GFP expression is under the control of the *Nur77* promoter (*Moran et al., 2011*). *Nur77* is an immediate early gene whose transcription is induced in lymphocytes by TCR or BCR engagement. Primary Nur77^GFP B cells were incubated with COS-7 cells expressing the single chain anti-Igκ surrogate Ag, or with parental COS-7 cells (no surrogate Ag), in the presence of CK-689 or CK-666. GFP expression was measured by flow cytometry after 3 hr (see *Figure 9—figure supplement 1* for gating strategy). In the absence of surrogate Ag expression by the COS-7 cells, Nur77^GFP B cells did not adhere to the COS-7 cells and GFP expression was not increased relative to B cells that were

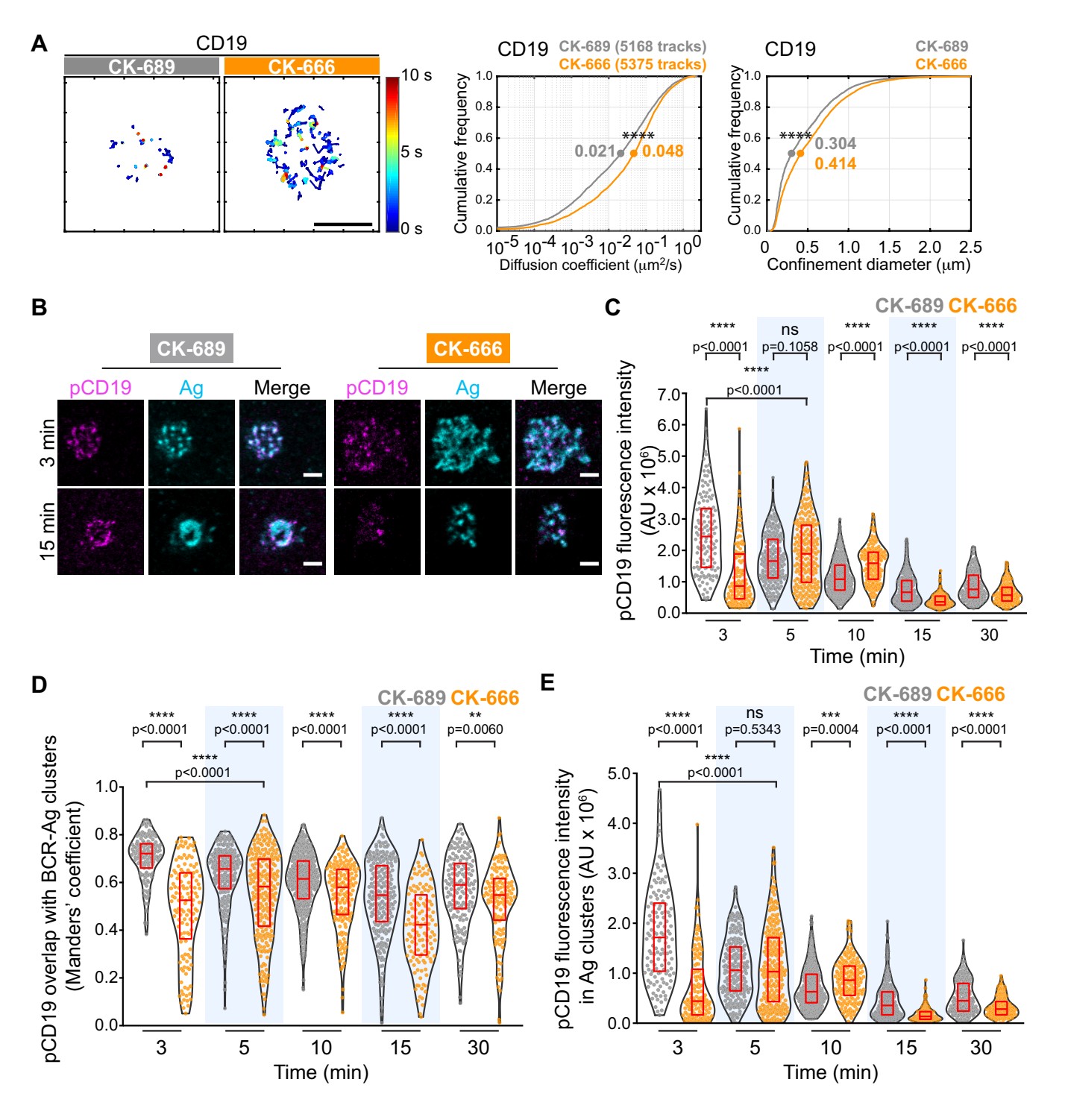

**Figure 8.** The Arp2/3 complex regulates CD19 mobility and BCR-CD19 interactions. (**A**) Ex vivo primary murine splenic B cells were treated with 100 µM CK-689 or CK-666 for 1 hr. SPT was then carried out as in *Figure 7*, using Cy3-labeled Fab fragments of antibodies to CD19. Single-particle trajectories from a representative cell are plotted using a color-coded temporal scale (left panels). Scale bars: 5 µm. Diffusion coefficients (center panels) and the diameter of maximum displacement (confinement diameter, right panels) over the 10 s period of observation were calculated for each track and cumulative frequency curves are shown. The dots on the curves indicate the median values. ****p<0.0001; Kolmogorov-Smirnov test. (**B–E**) Primary murine B cells were pre-treated with 100 µM CK-689 or CK-666 for 1 hr and then added to COS-7 cells expressing the single-chain anti-Igκ surrogate Ag. Cells were fixed at the indicated time points and stained with an antibody that recognizes the surrogate Ag and with an antibody that recognizes

*Figure 8 continued on next page*

*Figure 8 continued*
phosphorylated CD19. Representative cells are shown (**B**). Scale bars: 2 µm. For each B cell, the total fluorescence intensity of clustered pCD19 was calculated. Beeswarm plots in which each dot represents one cell are plotted with the median (red line) and interquartile ranges (red box) for >125 cells per time point from a representative experiment (**C**). For each cell in (**C**), the fraction of total pCD19 fluorescence that overlaps with BCR-Ag microclusters was quantified by calculating the Manders' coefficient (**D**). For each cell in (**C**), the total fluorescence intensity of pCD19 that was within BCR-Ag microclusters in cells was quantified (**E**). ****p<0.0001; ***p<0.001; ns, not significant; Mann-Whitney U test.
DOI: https://doi.org/10.7554/eLife.44574.031
The following figure supplement is available for figure 8:

**Figure supplement 1.** Arp2/3 complex activity increases CD19 phosphorylation in response to membrane-bound Ags.
DOI: https://doi.org/10.7554/eLife.44574.032

cultured in the absence of COS-7 cells (*Figure 9—figure supplement 1B,C*). By contrast, culturing Nur77[GFP] B cells with anti-Igκ-expressing COS-7 cells resulted in increased GFP expression compared to B cells that were cultured with parental COS-7 cells. Importantly, in the presence of the Arp2/3 inhibitor CK-666, this APC-induced increase in GFP expression was reduced to ~30% of the levels observed in control Nur77[GFP] B cells treated with CK-689 (*Figure 9B*).

In the presence of APC-bound Ags, inhibition of Arp2/3 complex activity also impaired the ability of primary B cells to increase their cell surface levels of CD69, the canonical marker of lymphocyte activation, and CD86, a co-stimulatory ligand critical for T cell activation (*Figure 9C,D*). Culturing primary B cells with anti-Igκ-expressing COS-7 APCs for 18 hr resulted in increased expression of CD69 and CD86 on the B cells (*Figure 9C,D*), responses that were completely dependent on expression of the surrogate Ag on the APCs (*Figure 9—figure supplement 2*). In CK-666-treated B cells, BCR-induced upregulation of CD69 and CD86 was inhibited by 40–50%. This was not due to increased B cell death, as DAPI-positive dead cells were excluded from analysis (*Figure 9—figure supplement 2*). Importantly, CD86 upregulation in response to PMA plus ionomycin was not affected by CK-666 (*Figure 9E*). Similar results were obtained for CD69 (data not shown). This suggests that inhibiting the Arp2/3 complex impairs proximal BCR signaling that is important for upregulation of CD86, as opposed to downstream events that are induced by both PMA plus ionomycin and BCR engagement.

In this co-culture system, both the B cells and the APCs are exposed to CK-666 for the entire duration of the experiment. To ensure that the observed reduction in CD69 and CD86 expression was not due to effects of CK-666 on the APCs, we used Arp3 siRNA to deplete the Arp2/3 complex in B16F1 murine melanoma cells expressing the single chain anti-Igκ Ag (*Figure 9—figure supplement 3A,B*). When primary B cells were added to these APCs, we observed that control and Arp3-depleted B16F1 APCs induced the upregulation of CD69 and CD86 to the same extent (*Figure 9—figure supplement 3C,D*). Moreover, B cells treated with CK-666 exhibited similarly impaired CD69 and CD86 responses, regardless of whether Arp3 was depleted or not in the APCs (*Figure 9—figure supplement 3E,F*). This indicates that the impaired upregulation of CD69 and CD86 observed in CK-666 treated cells is B cell-intrinsic.

Because the CK-666 was present during the entire B cell-APC co-culture in these experiments, it was also important to determine whether the impaired responses to APC-bound Ags were due to alterations in the initial BCR spatial reorganization and signaling, or to later events. To test this, we delayed the addition of CK-666. Pre-treating B cells with CK-666 for 1 hr prior to adding them to the APCs resulted in substantial inhibition of CD69 upregulation (*Figure 9C*). However, if the CK-666 was added 5 min after initiating the B cell-APC co-culture, or at later time points, it had very little effect on the BCR-induced upregulation of CD69 (*Figure 9F*). Adding CK-666 to the B cells at same time that they were mixed with the APCs caused only partial inhibition of the CD69 response compared to pre-treating the B cells for 1 hr with the drug. Similar results were obtained for CD86 (data not shown). This likely reflects the fact that some time is required for the CK-666 concentration in the B cells to reach a level that substantially inhibits Arp2/3 complex activity. Hence APC-induced B cell activation is only impaired if Arp2/3 complex activity is inhibited during the initial stages of B cell-APC interaction. Moreover, if CK-666 was present during the initial B cell-APC interaction, and then washed out after 5–120 min, CD69 upregulation was able to recover to nearly 100% of control values, regardless of when the washout was performed (*Figure 9—figure supplement 4A*). Once the drug was removed, the cells recovered their ability to aggregate BCR microclusters and form a

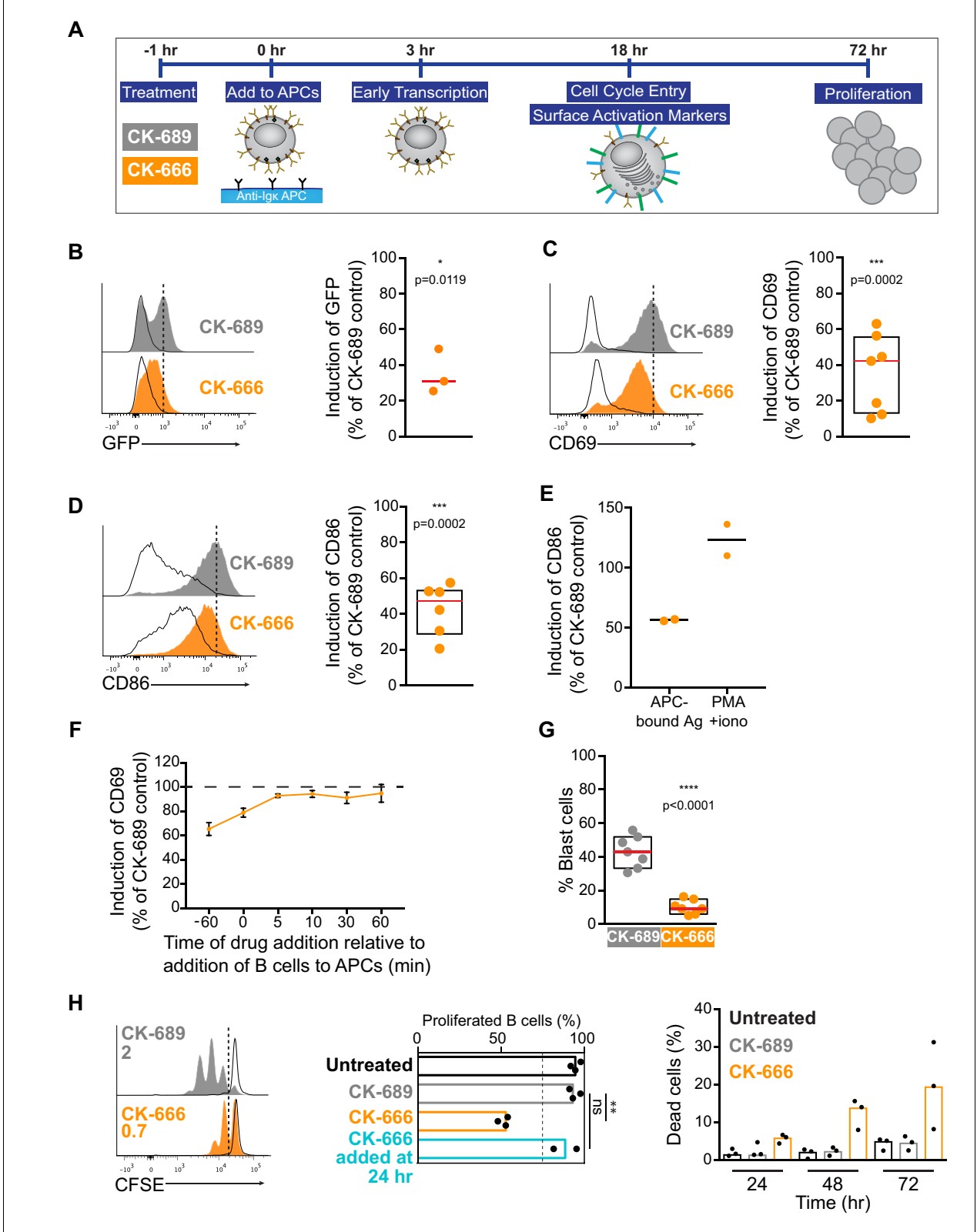

**Figure 9.** Arp2/3 complex activity is required for B cell activation responses. (**A**) Readouts used to assess B cell activation in response to anti-Igκ-expressing APCs. Primary murine B cells were pre-treated with CK-689 or CK-666 for 1 hr and then added to the APCs. The CK-689 or CK-666 was present in the co-culture for the entire length of the experiment unless otherwise indicated. (**B**) Histograms showing GFP fluorescence in primary ex vivo Nur77[GFP] B cells exposed to anti-Igκ-expressing COS-7 APCs (filled curves) or parental COS-7 cells (unfilled curves) for 3 hr. The Ag-induced increase in

*Figure 9 continued on next page*

*Figure 9 continued*

GFP fluorescence was calculated as the geometric mean for B cells cultured with anti-Igκ-expressing APCs (dotted line) minus the geometric mean for B cells cultured with parental COS-7 cells. See *Figure 9—figure supplement 1A* for gating strategy. The graph shows the Ag-induced increase in GFP fluorescence in CK-666-treated B cells as a percent of the response in CK-689-treated control cells (=100%). Each dot is an independent experiment and the red line is the median. (**C,D**) Histograms showing CD69 (**C**) or CD86 (**D**) upregulation in primary ex vivo C57BL/6J B cells exposed to anti-Igκ-expressing APCs (filled curves) or parental COS-7 cells (unfilled curves) for 18 hr. The Ag-induced increases in CD69 or CD86 expression were calculated as in (**B**). See *Figure 9—figure supplement 2* for gating strategy and representative calculations. Graphs show the increase in expression in CK-666-treated B cells as a percent of the response in CK-689-treated controls (=100%). Each dot is an independent experiment. The median (red line) and interquartile ranges (box) are shown. (**E**) Induction of CD86 expression in response to APCs (calculated as above) or to PMA +ionomycin (geometric mean for stimulated B cells minus geometric mean for unstimulated B cells) in the same experiment. Responses by CK-666-treated B cells are expressed as a percent of those in the CK-689-treated control cells. Results from two experiments are shown along with the average (bar). (**F**) B cells were pre-treated with CK-689 or CK-666 for 1 hr before being added to anti-Igκ-expressing APCs or parental COS-7 cells (−60 min time point). Alternatively, the drugs were added at the same time that the B cells (0 min) were added to the APCs or parental COS-7 cells, or at 5–60 min after initiating the co-culture. The Ag-induced increase in CD69 expression after 18 hr of co-culture was calculated as above and responses by CK-666-treated B cells are expressed as a percent of those in the CK-689-treated control cells. For each point, the average ± range is shown for two experiments. (**G**) B cells were pre-treated with CK-689 or CK-666 for 1 hr prior to being added to anti-Igκ-expressing APCs. After 18 hr of co-culture, the percent of blast cells with increased forward and side scatter was determined by flow cytometry (see *Figure 9—figure supplement 2* for gating strategy). Each dot is an independent experiment. The median (red line) and interquartile ranges are shown. (**H**) CFSE-labeled B cells were pre-treated with CK-689 or CK-666 for 1 hr prior to being cultured with APCs, IL-4, and BAFF for 3 days (filled curves). The unfilled curves depict CFSE dilution at Day 1. Representative data from one experiment is shown on the left and the average number of divisions per cell is indicated. The graph in the center shows the percent of live cells that had proliferated by Day 3. The percent of dead B cells that stained with 7-AAD is shown (right). Where indicated, CK-666 was added 24 hr after initiating the B cell-APC co-culture instead of 1 hr prior. In the graphs, each dot is an independent experiment and the bars indicate medians. ****p<0.0001; ***p<0.001; **p<0.01; *p<0.05; ns, not significant. Two-tailed paired t-test.

DOI: https://doi.org/10.7554/eLife.44574.033

The following figure supplements are available for figure 9:

**Figure supplement 1.** Gating strategy for GFP expression.
DOI: https://doi.org/10.7554/eLife.44574.034
**Figure supplement 2.** Gating strategy for CD69 and CD86 upregulation and blast cell formation.
DOI: https://doi.org/10.7554/eLife.44574.035
**Figure supplement 3.** Depletion of the Arp2/3 complex in APCs does not affect B cell activation responses.
DOI: https://doi.org/10.7554/eLife.44574.036
**Figure supplement 4.** Washout of CK-666 allows normal APC-induced upregulation of CD69 and cSMAC formation.
DOI: https://doi.org/10.7554/eLife.44574.037
**Figure supplement 5.** Inhibition of myosin does not affect upregulation of B cell activation markers or cSMAC formation.
DOI: https://doi.org/10.7554/eLife.44574.038
**Figure supplement 6.** Images of full blots.
DOI: https://doi.org/10.7554/eLife.44574.039

cSMAC (*Figure 9—figure supplement 4B–D*) and B cell activation could proceed normally, consistent with the effects of CK-666 on actin dynamics being rapidly reversible (*Yang et al., 2012*). Together, these results argue that Arp2/3 complex-dependent processes that occur during the initial stages of B cell-APC interactions are required for B cell activation responses.

Inhibition of the Arp2/3 complex results in dramatic alterations of the actin cytoskeleton at the immune synapse (*Figure 3*). It is possible that this disrupted actin architecture blocks contractile forces generated by myosin IIA (*Ennomani et al., 2016*). In T cells, contractile forces generated by myosin IIA are important for centralizing TCR microclusters into a cSMAC (*Babich et al., 2012*; *Ilani et al., 2009*; *Jacobelli et al., 2004*; *Kumari et al., 2012*; *Murugesan et al., 2016*; *Yi et al., 2012*). To assess the contribution of myosin contractility in our system, we treated primary B cells with para-nitro-blebbistatin (pnBB) to inhibit myosin. When pnBB-treated cells were added to COS-7 APCs expressing membrane-bound anti-Igκ for 18 hr, they exhibited a slight, but not statistically significant reduction in CD69 and CD86 upregulation compared to DMSO-treated cells (*Figure 9—figure supplement 5A,B*). Moreover, pnBB-treated A20 D1.3 B cells that were added to COS-7 APCs expressing mHEL-HaloTag were able to form cSMACs to the same extent as DMSO-treated cells (*Figure 9—figure supplement 5A,B*). These results suggest that APC-induced activation marker expression and cSMAC formation are dependent on Arp2/3 complex activity, whereas myosin contractility has little or no involvement, at least under these experimental conditions.

As part of their activation program, B cells increase in size as they enter S phase, reflecting an increase in the secretory machinery required for antibody synthesis (*DeFranco, 1985*; *Kirk et al., 2010*). Compared to resting lymphocytes, activated lymphoblasts exhibit an increase in both forward scatter (cell size) and side scatter (granularity) when analyzed by flow cytometry. Co-culturing primary B cells with anti-Igκ-expressing COS-7 APCs for 18 hr resulted in 30–55% of the B cells becoming larger, more granular blast cells (*Figure 9G*; *Figure 9—figure supplement 2C,D*). However, in the presence of the Arp2/3 complex inhibitor CK-666, the percent of B cells that became blast cells was reduced to 5–16% (*Figure 9G*).

In the presence of co-stimulatory cytokines such as BAFF and IL-4, Ag-activated B cells proliferate extensively. CFSE dilution assays showed that >90% of primary B cells that were co-cultured with anti-Igκ-expressing COS-7 APCs plus these cytokines for 72 hr had divided (*Figure 9H*). Addition of CK-666 to the culture reduced the percent of B cells that had divided at least once to ~55% and the mean number of cell divisions was reduced from 2 to 0.7 (*Figure 9H*). This reduction in B cell proliferation was accompanied by increased cell death (*Figure 9H*), which is the alternative fate for B cells that are cultured in vitro in the absence of mitogenic stimuli. Note that when the Arp2/3 inhibitor CK-666 was added 24 hr after initiating the B cell-APC co-culture, B cell proliferation was not impaired (*Figure 9H*, middle panel). This indicates that CK-666 does not block cell division and suggests that the critical functions of the Arp2/3 complex in APC-dependent B cell activation occur during the first 24 hr. Collectively, these findings indicate that the Arp2/3 complex-dependent processes are important for B cells to enter S phase and proliferate.

## Discussion

Using high spatiotemporal resolution imaging of actin in B cells interacting with APCs, we observed that B cells scan the surface of the APC using complex and dynamic actin-rich protrusions that constantly extend and retract across the APC surface. This dynamic probing is mediated by Arp2/3 complex activity, which also generates retrograde actin flow that facilitates the initial centripetal movement of BCR microclusters and is required for cSMAC formation. Importantly, these Arp2/3 complex-dependent processes amplify proximal and distal BCR signaling, transcriptional responses, and B cell proliferation in response to APC-bound Ag.

In contrast to the highly dynamic lamellipodia and filopodia that B cells extend asymmetrically over the APC surface, the actin-rich protrusions that B and T cells form when they spread on Ag-bearing lipid bilayers (*Fleire et al., 2006*; *Murugesan et al., 2016*) or Ag-coated coverslips (*Bunnell et al., 2001*; *Freeman et al., 2011*) are predominantly symmetrical, sheet-like lamellipodia. This difference in spreading and probing behavior may reflect the physical properties of the Ag-bearing substrate. In fibroblasts and epithelial cells, stiffer substrates favor Rac1- and Arp2/3 complex-dependent formation of lamellipodia, whereas softer substrates favor Cdc42- and formin-dependent formation of filopodia (*Collins et al., 2017*; *Wong et al., 2014*). The greater prevalence of filopodia when B cells spread on APCs, compared to artificial lipid bilayers, is consistent with the finding that the APC membrane is less stiff than the artificial lipid bilayers that have been used to mimic APCs (*Natkanski et al., 2013*). Whether local variations in the stiffness of the APC membrane shape B cell probing behavior is not known. However, the membranes of FDCs and dendritic cells differ in their stiffness (*Spillane and Tolar, 2017*) and this could impact the type of probing behavior that B cells use to scan the surface of these APCs.

The filopodia-like structures formed by control B cells interacting with APCs allowed the formation of BCR-Ag microclusters at sites distant from the cell body. Branched actin networks that were nucleated around these linear protrusions then encased the BCR-Ag microclusters and generated centripetal flow that was required for BCR-Ag microcluster centralization. When the Arp2/3 complex was inhibited, lamellipodial branched actin protrusions were not generated in response to either APC-bound Ag or anti-Ig-coated coverslips, and instead the formation of filopodia was greatly enhanced. This is consistent with the idea that the Arp2/3 complex and formins compete for a limited pool of actin monomers (*Suarez and Kovar, 2016*). In B cells in which the Arp2/3 complex was inhibited, the filopodia that formed were able to extend over the surface of the APC and gather Ag into microclusters. However, these filopodia and their associated BCR-Ag microclusters, were not efficiently retracted toward the cell body.

We showed that Arp2/3 complex-dependent branched actin nucleation is essential for actin retrograde flow at the cell periphery and is required for BCR-Ag microcluster centralization and cSMAC formation. In T cells, both pushing forces generated by actin retrograde flow and contractile forces generated by actomyosin arcs have been shown to drive the centripetal movement of TCR microclusters to form a cSMAC (*Babich et al., 2012*; *Ilani et al., 2009*; *Jacobelli et al., 2004*; *Kumari et al., 2012*; *Murugesan et al., 2016*; *Yi et al., 2012*). In B cells spreading on anti-Ig-coated coverslips, we observed that similar actomyosin arcs formed at the inner face of the peripheral ring of branched actin. Consistent with the idea that these linear actin arcs are generated by formins, these structures appeared to form in B cells in which the Arp2/3 complex was inhibited, as has been observed in T cells (*Murugesan et al., 2016*). Myosin IIA is important for B cells to extract Ags from APC membranes (*Hoogeboom et al., 2018*; *Natkanski et al., 2013*) and has been implicated in BCR microcluster centralization (*Tolar, 2017*). However, inhibition of myosin IIA did not affect cSMAC formation or the upregulation of B activation markers in our system. Although this does not rule out a role for myosin in B cell immune synapse formation and APC-induced B cell activation, it highlights that the Arp2/3 complex has a more important role in these processes, at least in our experimental system.

TCR microcluster centralization and cSMAC formation in T cells has been proposed to involve three cytoskeleton-dependent mechanisms that act sequentially: actin retrograde flow, actomyosin contraction, and movement along microtubules that span the actin-poor central region of the immune synapse (*Kumari et al., 2014*). Dynein-dependent movement of BCR and TCR microclusters along juxtamembrane microtubules is required for cSMAC formation in both B and T cells (*Hashimoto-Tane et al., 2011*; *Schnyder et al., 2011*). We have shown that the formation of branched actin at the periphery of the immune synapse is required for capturing the plus ends of microtubules and for moving the microtubule network toward the Ag-presenting surface (*Wang et al., 2017*). In B cells spreading on anti-Ig-coated coverslips, microtubule plus end binding proteins contact the inner face of the dense peripheral actin ring, which is depleted of microtubules, much like the leading edge of a migrating cell. Hence, initial microcluster centralization mediated by Arp2/3 complex-dependent actin retrograde flow may be required for BCR microclusters to associate with microtubules and coalesce into a cSMAC at the center of the immune synapse. Thus, inhibiting Arp2/3 complex activity may block a critical initial step in BCR-Ag microcluster centralization and prevent the microtubule-based centralization that leads to cSMAC formation, the site where Ags are often internalized.

We showed that Arp2/3 complex activity amplifies BCR signaling in response to APC-bound Ags. The molecular mechanisms by which the actin cytoskeleton regulates Ag receptor signaling are still being explored. In T cells, actin retrograde flow is important for sustained TCR signaling in response to Ag-bearing artificial lipid bilayers (*Babich et al., 2012*; *Kumari et al., 2015*). Because the TCR is a mechanosensitive receptor, actin retrograde flow could exert tension on the TCR that enhances its signaling (*Basu and Huse, 2017*). The BCR is also highly sensitive to mechanical force, such that increasing the stiffness of Ag-presenting surfaces causes greater recruitment of Syk and other signaling components (*Wan et al., 2013*; *Wan et al., 2015*). Hence, Arp2/3 complex-dependent actin retrograde flow could increase the mechanical tension on Ag-bound BCRs and enhance their signaling. Consistent with this idea, we observed that peak BCR signaling occurred during the first 5 min of APC encounter, when BCR-Ag microclusters are undergoing the greatest centripetal movement.

BCR signaling is enhanced by the formation of large and stable microclusters, which is dependent on Ag mobility and the actin cytoskeleton (*Ketchum et al., 2014*). In control B cells, microcluster size increased most rapidly between 2 and 6 min after the B cells were added to the APCs, which corresponds with the peak of proximal BCR signaling. Inhibition of the Arp2/3 complex decreased both microcluster merge events and BCR signaling, consistent with the idea that actin-dependent microcluster aggregation enhances BCR signaling. Microcluster growth may increase the frequency at which Ag-bound BCRs that have recruited Syk can trap BCRs that are not bound to Ag, and activate them via Syk-mediated phosphorylation. Alternatively, larger microclusters may more efficiently shield activated BCRs from negative regulators such as CD22 (*Gasparrini et al., 2016*).

Arp2/3 complex-dependent actin polymerization could also enhance BCR signaling by maintaining the integrity of the BCR-Ag microclusters and by acting as a platform for the assembly of signaling complexes. We observed that BCR-Ag microclusters were surrounded by actin cages. The idea that these cages maintain the integrity of BCR microclusters is supported by the finding that

microclusters become more diffuse when B cells are treated with the actin-depolymerizing drug latrunculin A (*Treanor et al., 2011*). In T cells, signaling proteins such as PLCγ1 accumulate at actin foci that surround TCR microclusters (*Kumari et al., 2015*). Deletion of WASp or inhibition of the Arp2/3 complex has no effect on proximal TCR signaling or microcluster formation but results in impaired recruitment and activation of PLCγ1 (*Kumari et al., 2015*). Whether Arp2/3 complex-dependent actin structures surrounding BCR microclusters act as scaffolds that amplify and diversify BCR signaling remains to be determined.

Although Ag-induced actin reorganization enhances BCR signaling, the submembrane actin cytoskeleton limits the lateral mobility of the BCR in resting B cells and thereby prevents spontaneous BCR signaling. Indeed, disrupting the actin cytoskeleton with latrunculin A is sufficient to induce robust Ag-independent BCR signaling (*Treanor et al., 2010*). We found that inhibiting Arp2/3 complex activity in resting primary B cells resulted in increased basal levels of pERK and pAkt, and that this correlated with increased lateral mobility of both IgM-BCRs and IgD-BCRs within the plasma membrane. Hence, Arp2/3 complex-mediated actin polymerization is also important for maintaining the quiescent state of resting B cells and for preventing aberrant Ag-independent BCR signaling.

We also showed that Arp2/3 complex activity enhances functional interactions between the BCR and its co-receptor CD19. Ag binding promotes the tyrosine phosphorylation of CD19, allowing it to recruit key effectors of BCR signaling such as PI3K and Vav. CD19 associates with the Lyn tyrosine kinase, which can phosphorylate the CD79a/b signaling subunit of the BCR, and has also been reported to enhance recruitment of the Syk and Btk tyrosine kinases to the BCR (*Depoil et al., 2008*; *Fujimoto et al., 2000*; *Fujimoto et al., 2002*). Hence, the interaction of the BCR with CD19 facilitates processive signaling reactions that amplify BCR signaling. We found that inhibiting the Arp2/3 complex resulted in decreased CD19 phosphorylation, as well as reduced overlap between BCR-Ag clusters and pCD19 clusters, at 3 min after the initiating B cell-APC contact. By 5 min, the differences between CK-666-treated B cells and control cells were smaller. Hence, inhibiting the Arp2/3 complex reduces and delays initial Ag-induced BCR-CD19 interactions. Initial BCR-CD19 interactions are dependent on the immobilization of CD19 by the tetraspanin network (*Mattila et al., 2013*). We found that inhibiting Arp2/3 complex-dependent actin polymerization increased the lateral mobility of CD19 within the plasma membrane, suggesting that branched actin structures also restrain its mobility. This increased mobility of CD19 may decrease the initial rate at which productive BCR-CD19 collisions occur.

Although inhibiting the Arp2/3 complex significantly reduced proximal BCR signaling during the first 5 min of B cell-APC encounter, a key question was whether this would translate into decreased B cell activation. Indeed, CK-666 treatment reduced the ability of Ag-bearing APCs to stimulate a transcriptional response at 3 hr, activation marker upregulation at 18 hr, and B cell proliferation at 72 hr. Importantly, we found that adding CK-666 to B cells shortly after they had engaged APCs did not impair the upregulation of B cell activation markers. As well, we found that Arp2/3 complex activity was not required for the upregulation of activation markers in response to PMA plus ionomycin, stimuli that bypass the BCR but initiate many of the same downstream signaling reactions. Taken together, these findings support the idea that Arp2/3 complex-dependent effects on BCR-Ag microclusters that occur within the first few minutes of B cell-APC interactions amplify initial BCR signaling reactions that are translated into B cell responses that occur up to 3 days later.

Consistent with our findings that Arp2/3 complex activity regulates both tonic BCR signaling and APC-induced B cell activation, upstream activators of the Arp2/3 complex are important regulators of B cell activation. Murine B cells lacking WASp, an NPF that activates the Arp2/3 complex, exhibit reduced cell spreading, BCR-Ag microcluster formation, and BCR-induced tyrosine phosphorylation in response to Ag-bearing lipid bilayers (*Liu et al., 2011*). The WASp-interacting protein WIP regulates the activity and subcellular distribution of WASp, and also stabilizes existing actin filaments. Murine B cells lacking WIP have defective CD19-mediated PI3K signaling and antibody responses (*Keppler et al., 2015*). WASp activation is regulated by the Cdc42 GTPase and B-cell-specific loss of Cdc42 results in aberrant actin organization, diminished BCR signaling, and a severe impairment in antibody production (*Burbage et al., 2015*).

In line with its central role in many aspects of B cell function, human mutations that impair the activation Arp2/3 complex result in disease. Mutations in WASp or WIP result in Wiskott-Aldrich syndrome, an X-linked immunodeficiency disorder (*Candotti, 2018*). Similarly, missense mutations in the *ARPC1B* gene have recently been described as causing a Wiskott-Aldrich syndrome-like disease

(*Kahr et al., 2017*). Although Wiskott-Aldrich syndrome is associated with increased susceptibility to infections, many patients also develop autoimmunity and B-cell malignancies (*Candotti, 2018*). Our finding that the loss of Arp2/3 complex activity in B cells results in increased tonic BCR signaling but impaired APC-induced B cell activation may help explain this paradox.

# Materials and methods

Key resources table

| Reagent type | Designation | Source or reference | Identifiers | Additional information |
|---|---|---|---|---|
| Sequence-based reagent | ON-TARGETplus Non-Targeting Pool, | Dharmacon | #D-001810-01-05 | |
| Sequence-based reagent | SMARTpool ON-TARGETplus, Dharmacon, | Dharmacon | #L-046642-01-0005 | |
| Other | RPMI-1640 without phenol red | Life Technologies | #32404014 | |
| Antibody | Goat anti-mouse IgG | Jackson ImmunoResearch | #115-005-008 | Coverslip coating 2.5 $\mu$g/cm$^2$ |
| Antibody | Goat anti-mouse IgM | Jackson ImmunoResearch | #115-005-020 | Coverslip coating 2.5 $\mu$g/cm$^2$ |
| Antibody | Rabbit anti-pZap70 (Y319)/pSyk(Y352) | Cell Signaling Technologies | #2701 | Immunofluorescence 1:200 |
| Antibody | Rabbit anti-pCD79a(Y182) | Cell Signaling Technologies | #5173 | Immunofluorescence 1:400, Western blot 1:1000 |
| Antibody | Rabbit anti-CD79a | *Gold et al., 1991* | | Western blot 1:5000 |
| Antibody | Rabbit anti-pCD19(Y531) | Cell Signaling Technologies | #3571 | Immunofluorescence 1:200, Western blot 1:1000 |
| Antiboday | Rabbit anti-CD19 | Cell Signaling Technologies | #3574 | Western blot 1:1000 |
| Antibody | Alexa Fluor 488 goat anti-rabbit IgG | Invitrogen | #A-11008 | Immunofluorescence 1:400 |
| Antibody | Alexa Fluor 647 goat anti-rat IgG | Invitrogen | #A-21247 | Immunofluorescence 1:400 |
| Antibody | Goat anti-mouse Ig kappa | Southern Biotech | #1050–01 | Stimulation 20 $\mu$g/ml |
| Antibody | Rabbit anti-ACTR3 (Arp3) | Santa Cruz | #sc-15390 | Western blot 1:1000 |
| Antibody | Rabbit anti-Arp2 | Abcam | #ab128934 | Western blot 1:1000 |
| Antibody | Mouse anti-$\beta$-actin | Santa Cruz | #sc-47778 | Western blot 1:5000 |
| Antibody | Rabbit anti-p34-Arc/ARPC2 | Millipore | #07–227 | Western blot 1:1000 |
| Antibody | Rabbit anti-pERK | Cell Signaling Technologies | #9101 | Western blot 1:1000 |
| Antibody | Rabbit anti-ERK | Cell Signaling Technologies | #9102 | Western blot 1:1000 |
| Antibody | Rabbit anti-pAkt | Cell Signaling Technologies | #9271 | Western blot 1:1000 |
| Antibody | Rabbit anti-Akt | Cell Signaling Technologies | #9272 | Western blot 1:1000 |
| Antibody | Goat anti-rabbit IgG(H + L)-HRP conjugate | Bio-Rad | #170–6515 | Western blot 1:3000 |
| Antibody | Goat anti-mouse IgM Fab fragment-Cy3-conjugated | Jackson ImmunoResearch | #115-167-020 | Single particle tracking 1 ng/ml |
| Antibody | Rat anti-mouse IgD Fab fragment-Cy3-conjugated | ATCC | #HB-250 (11–26 c) | Fab made by AbLab; SPT 100 ng/ml |

*Continued on next page*

Continued

| Reagent type | Designation | Source or reference | Identifiers | Additional information |
|---|---|---|---|---|
| Antibody | Rat anti-mouse CD19 Fab fragment-Cy3-conjugated | ATCC | #HB-305 (1D3) | Fab made by AbLab; SPT 10 ng/ml |
| Antibody | Rabbit anti-MHC II monoclonal antibody (M5/114) | Millipore | #MABF33 | Coverslip coating 0.25 µg/cm$^2$ |
| Antibody | 2.4G2 monoclonal antibody | ATCC | #HB-197 | 0.25 µg/ml |
| Antibody | Anti-CD69-FITC | eBioscience | #11-0691-82 | Flow cytometry 1:200 |
| Antibody | Anti-CD86-APC | eBioscience | #17-0862-82 | Flow cytometry 1:200 |
| Biological sample (bovine) | Fibronectin | Sigma-Aldrich | #F4759 | Coverslip coating 5 µg/ml |
| Cell line (C. aethiops) | COS-7 | American Type Culture Collection | #CRL-1651; RRID:CVCL_0224 | |
| Cell line (M. musculus) | A20 | American Type Culture Collection | #TIB-208; RRID:CVCL_1940 | |
| Cell line (M. musculus) | A20 D1.3 | *Batista and Neuberger, 1998* | | F Batista (Ragon Institute, Cambridge, MA) |
| Cell line (M. musculus) | B16F1 | American Type Culture Collection | #CRL-6323; RRID:CVCL_0158 | |
| Chemical compound, drug | CK-666 | Calbiochem | #CAS 442633-00-3 | 100 µM |
| Chemical compound, drug | CK-689 | Calbiochem | #CAS 170930-46-8 | 100 µM |
| Chemical compound, drug | (S)-nitro-blebbistatin (pnBB) | Cayman Chemicals | #CAS 856925-71-8 | 50 µM |
| Chemical compound, drug | HaloTag Janelia Fluor 549 ligand | Promega | #GA1110 | Cell labeling 1:2000 |
| Chemical compound, drug | HaloTag tetramethylrhodamine ligand | Promega | #G8251 | Cell labeling 1:2000 |
| Chemical compound, drug | Rhodamine phalloidin | Thermo Fisher | #R415 | Immunofluorescence 1:1000 |
| Chemical compound, drug | Alexa Fluor 568-conjugated phalloidin | Thermo Fisher | #A12380 | Immunofluorescence 1:1000 |
| Chemical compound, drug | Alexa Fluor 532-conjugated-phalloidin | Thermo Fisher | #A22282 | Immunofluorescence 1:1000 |
| Chemical compound, drug | ProLong Diamond anti-fade reagent | Thermo Fisher | #P36965 | |
| Chemical compound, drug | ECL detection reagent | GE Life Sciences | #RPN2106 | |
| Chemical compound, drug | CFSE | Invitrogen | #C1157 | Flow cytometry 2 µM |
| Chemical compound, drug | 7-AAD | Thermo Fisher | #A1310 | Flow cytometry 1:1000 |
| Commercial assay or kit | B cell isolation kit | Stemcell Technologies | #19854A | |
| Commercial assay or kit | AMAXA nucleofector kit V | Lonza | #VCA-1003 | |
| Commercial assay or kit | Ingenio electroporation solution | Mirus | #MIR 50117 | |
| Commercial assay or kit | Lipofectamine 3000 | Thermo-Fisher | #L3000008 | |

*Continued*

| Reagent type | Designation | Source or reference | Identifiers | Additional information |
|---|---|---|---|---|
| Commercial assay or kit | Fab preparation kit | Pierce | #44985 | |
| Genetic reagent (M. musculus) | C57BL/6J | Jackson Laboratories | Stock #000664; MGI:3028467 | |
| Genetic reagent (M. musculus) | C57BL/6-Tg (IghelMD4)4Ccg/J | Jackson Laboratories | Stock #002595; MGI:2384162 | *Goodnow et al., 1988* |
| Genetic reagent (M. musculus) | C57BL/6-Tg (Nr4a1-EGFP/cre) 820Khog | *Moran et al., 2011* | MGI:5007644 | PMID:21606508; K Hogquist (University of Minnesota) |
| Peptide, recombinant protein | BAFF | R&D Systems | #8876-BF-010 | 5 ng/ml |
| Peptide, recombinant protein | IL-4 | R&D Systems | #404 ML-010 | 5 ng/ml |
| Recombinant DNA reagent | F-tractin-GFP | *Johnson and Schell, 2009* | | |
| Recombinant DNA reagent | Myosin IIA-GFP | Addgene | #38297 | |
| Recombinant DNA reagent | mHEL-GFP | *Batista et al., 2001* | | F Batista (Ragon Institute, Cambridge, MA) |
| Recombinant DNA reagent | mHEL-HaloTag | PMID: 29388105 | | M Gold (University of British Columbia) |
| Recombinant DNA reagent | Single-chain anti-mouse Ig kappa antibody | *Ait-Azzouzene et al., 2005* | | D Nemazee (Scripps Reseach Institute) |
| Recombinant DNA reagent | β-Actin-GFP | | | R Nabi (University of British Columbia) |
| Software, algorithm | ISIM deconvolution | PMID: 26210400 | | |
| Software, algorithm | Softworx | Applided Precision Ltd.; GE Healthcare Life Sciences | | |
| Software, algorithm | FIJI | *Schindelin et al., 2012* | | https://imagej.net/Fiji |
| Software, algorithm | MATLAB | MathWorks | R2015b | |
| Software, algorithm | MetaMorph | Molecular Devices | | |
| Software, algorithm | ICY | *de Chaumont et al., 2012* | | http://icy.bioimage analysis.org |
| Software, algorithm | FlowJo | Tree Star | v10.5.3 | |
| Software, algorithm | Robust Regression and Outlier Removal (ROUT) | Graphad | v7 | *Motulsky and Brown, 2006* |

## B cells

Spleens were obtained from 8- to 12 week old C57BL/6J (Jackson Laboratories, #000664), Nur77[GFP](*Moran et al., 2011*), or MD4 (*Goodnow et al., 1988*) (Jackson Laboratories, #002595) mice of either sex following protocols approved by the University of British Columbia Animal Care Committee. B cells were isolated using a negative selection B cell isolation kit (Stemcell Technologies, #19854A). The A20 murine IgG[+] B-cell line (confirmed to be IgG[+] by flow cytometry and responsive to anti-mouse IgG antibodies) was obtained from ATCC (#TIB-208). A20 D1.3 B cells

expressing a transgenic HEL-specific BCR (confirmed to express both mouse IgG and IgM (the transfected D1.3 BCR) and to be responsive to HEL) were a gift from F. Batista (Ragon Institute, Cambridge, MA) (*Batista and Neuberger, 1998*). All cell lines were confirmed to be free of mycoplasma contamination. A20 B cells were cultured in RMPI-1640 supplemented with 5% fetal calf serum (FCS), 2 mM glutamine, 1 mM pyruvate, 50 µM 2-mercaptoethanol, 50 U/ml penicillin, and 50 µg/ml streptomycin (complete medium). A20 and A20 D1.3 B cells ($3 \times 10^6$ cells) were transiently transfected using the AMAXA nucleofector kit V (Lonza, #VCA-1003) or the Ingenio electroporation kit (Mirus, #MIR 50117) with 1.5 µg of either control siRNA (ON-TARGETplus Non-Targeting Pool, Dharmacon, #D-001810-01-05) or Actr3 siRNA (SMARTpool ON-TARGETplus, Dharmacon, #L-046642-01-0005), or with 2 µg of plasmid DNA encoding either F-tractin-GFP (*Johnson and Schell, 2009*) or myosin IIA-GFP (Addgene, #38297) (*Jacobelli et al., 2009*). Transfected cells were cultured for 24 hr before use. Where indicated, B cells ($7.5 \times 10^5$ cells/ml) were pre-treated with 100 µM of the Arp2/3 complex inhibitor CK-666 (Calbiochem, #CAS 442633-00-3) or the inactive analog CK-689 (Calbiochem, #CAS 170930-46-8) (*Hetrick et al., 2013*) for 1 hr at 37°C or with 50 µM of the myosin IIA inhibitor pnBB (Cayman Chemicals, #CAS 856925-71-8) or DMSO for 30 min at 37°C.

## APCs

APCs were generated as described previously (*Freeman et al., 2011*; *Wang et al., 2017*; *Wang et al., 2018*). Briefly, COS-7 cells (ATCC, #CRL-1651) or B16F1 cells (ATCC, #CRL-6323) were transiently transfected using Lipofectamine 3000 (Thermo Fisher, #L3000008) with plasmid DNA encoding either mHEL-GFP, mHEL-HaloTag, or a single-chain anti-Igκ antibody. mHEL-GFP is a transmembrane protein consisting of the complete HEL and GFP proteins in the extracellular domain, fused to the transmembrane region and cytosolic domain of the H-2K$^b$ protein (*Batista et al., 2001*). mHEL-HaloTag consists of the complete HEL protein fused to the transmembrane region and cytosolic domain of H-2K$^b$, with the the HaloTag protein fused to the C-terminus (see *Figure 1*) (*Wang et al., 2018*). The single-chain anti-Igκ antibody is a transmembrane protein consisting of the single-chain Fv with the variable regions from the 187.1 rat anti-Igκ monocolonal antibody, fused to the hinge and membrane-proximal domains of rat IgG1 and the transmembrane and cytoplasmic domains of the H-2K$^b$ protein (*Ait-Azzouzene et al., 2005*). Where indicated, B16F1 cells were cotransfected with 1.5 µg of either control siRNA (ON-TARGETplus Non-Targeting Pool, Dharmacon, #D-001810-01-05) or Actr3 siRNA (SMARTpool ON-TARGETplus, Dharmacon, #L-046642-01-0005). The transfected cells were cultured in DMEM supplemented with 5% FCS, 2 mM glutamine, 1 mM pyruvate, 50 U/ml penicillin, and 50 µg/ml streptomycin for 24 hr before being used for experiments.

## Fluorescence recovery after photobleaching

COS-7 cells that had been transfected with actin-GFP ($1.5 \times 10^5$ cells) were plated onto fibronectin-coated coverslips and cultured overnight. Fluorescence recovery after photobleaching was performed as described previously (*Freeman et al., 2011*) using an Olympus FV1000 confocal microscope with a 100X NA 1.40 oil objective. Briefly, within a selected region of interest (ROI), the pre-bleached fluorescence intensity was measured, followed by photobleaching using the 405 nm laser (100% intensity for 3 s). Fluorescence recovery within the same ROI was then recorded for 40 s. Fluo-View v4.0 (Olympus) software was used to quantify the fluorescence within the ROI, which was then normalized to the pre-bleach fluorescence intensity.

## BCR microcluster dynamics

Live cell imaging of BCR-Ag microcluster dynamics was performed as described previously (*Wang et al., 2018*). mHEL-GFP-expressing COS-7 cells were plated onto fibronectin-coated coverslips and cultured overnight. The coverslip was then washed with modified HEPES-buffered saline (mHBS; 25 mM HEPES, pH 7.2, 125 mM NaCl, 5 mM KCl, 1 mM CaCl$_2$, 1 mM Na$_2$HPO$_4$, 0.5 mM MgSO$_4$, 1 mg/ml glucose, 2 mM glutamine, 1 mM pyruvate, 50 µM 2-mercaptoethanol) supplemented with 2% FCS (imaging medium) and mounted onto the microscope in a 37°C imaging chamber. A20 D1.3 B cells were pre-treated with CK-689 or CK-666 for 1 hr in imaging medium. The B cells ($5 \times 10^5$ in 0.1 ml) were then added to the COS-7 APCs and imaging was started immediately. Imaging was performed using a spinning disk confocal microscope system (Intelligent Imaging Innovations)

consisting of a Zeiss Axiovert 200M microscope with a 100X NA 1.45 oil Pan-Fluor objective and a QuantEM 512SC Photometrics camera. Z-stacks with 0.2-µm optical z-slices through the B cell-APC contact site were acquired at 12 s intervals. Alternatively, F-tractin-GFP-expressing A20 D1.3 B cells were pre-treated with CK-689 or CK-666, added to mHEL-HaloTag-expressing COS-7 cell APCs that had been labeled with the Janelia Fluor 549 dye (*Grimm et al., 2015*), and imaged using 3D-ISIM (*York et al., 2013*). Z-stacks through the B cell-APC contact site were acquired every 6.6 s and the images were deconvolved as described (*York et al., 2013*). Briefly, the images were subjected to 10 iterations of Richardson-Lucy deconvolution with a Gaussian point spread function having a full width half maximum (0.22 µm lateral and 0.5 µm axial) that was derived from measurements of 100 nm yellow/green beads. Within a single experiment, all images were acquired using identical settings. FIJI software (*Schindelin et al., 2012*) was used to quantify cell area and generate kymographs. Custom FIJI macros were used to quantify the fluorescence intensity of Ag in microclusters and microcluster size (https://github.com/madscience12/FIJImacros/blob/master/APC_analyzer_MBM.ijm) as well as the distance between each microcluster and the center of the immune synapse (https://github.com/madscience12/FIJImacros/blob/master/distance_macro_MBM.ijm; *Bolger-Munro, 2019*).

## Actin organization and dynamics in B cells spreading on anti-Ig-coated coverslips

Glass coverslips were coated with 2.5 µg/cm$^2$ goat anti-mouse IgG (for A20 B cells; Jackson ImmunoResearch, #115-005-008) or goat anti-mouse IgM (for primary B cells; Jackson ImmunoResearch, #115-005-020) as described (*Lin et al., 2008*). B cells were pre-treated for 1 hr with CK-666 or CK-689 before adding $7.5 \times 10^4$ cells in 0.1 ml imaging medium to the coverslips. Cells were fixed and permeabilized as described below for immunostaining. F-actin was visualized using rhodamine- or Alexa Fluor 568-conjugated phalloidin (Thermo Fisher, #R415, #A12380, respectively; 1:1000). For live cell imaging at 37°C, F-tractin-GFP-expressing A20 B cells were imaged using either a Zeiss LSM 880 Airyscan microscope in super-resolution mode with a 63X 1.4 NA objective or a TIRF-SIM microscope (DeltaVision OMX-SR, GE Healthcare Life Sciences) equipped with a 60X 1.42 NA objective (Olympus). For Airyscan images, raw data were processed using automatically determined parameters. For TIRF-SIM images, raw data were reconstructed using Softworx (Applied Precision Ltd.; GE Healthcare Life Sciences) and with a value of 0.01 for the Wiener filter constant. Alternatively, the cells were fixed at the indicated times with 4% paraformaldehyde (PFA), permeabilized, and stained for F-actin as described below. FIJI software (*Schindelin et al., 2012*) was used to quantify cell area and generate kymographs.

## STED microscopy

STED microscopy was performed as described previously (*Wang et al., 2018*). Briefly, $5 \times 10^4$ A20 B cells in 0.1 ml imaging medium were allowed to spread on anti-IgG-coated coverslips before being fixed and stained with Alexa Fluor 532-conjugated-phalloidin (Thermo Fisher, #A22282). STED images were acquired using a Leica TCS SP8 laser scanning STED system equipped with a 592 nm depletion laser, a CX PL APO 100X NA 1.40 oil objective, and a Leica HyD high sensitivity detector. Image deconvolution was performed using Huygens software (Scientific Volume Imaging, Hilversum, Netherlands).

## Analysis of APC-induced BCR signaling by immunostaining

Primary murine B cells ($10^5$ in 0.1 ml imaging medium) from C57BL/6J or MD4 mice that had been pre-treated with CK-689 or CK-666 were added to COS-7 APCs expressing either the single-chain anti-Igκ surrogate Ag or mHEL-HaloTag that was labeled with HaloTag tetramethylrhodamine ligand (Promega, #G8251). After the indicated times at 37°C, the cells were fixed with 4% PFA for 10 min at room temperature. The cells were then permeabilized with 0.1% Triton X-100 in PBS for 3 min and blocked with 2% BSA in PBS for 30 min, both at room temperature. Primary antibodies that were diluted in blocking buffer (2% BSA in PBS) were added for 1 hr at room temperature, or overnight at 4°C for anti-pCD79. The primary antibodies, all from Cell Signaling Technologies, were: pZap70 (Y319)/pSyk(Y352) (#2701; 1:200), pCD79a(Y182) (#5173; 1:400), and pCD19(Y531) (#3571; 1:200). The coverslips were then incubated for 30 min at room temperature with a mixture of Alexa Fluor 488 goat anti-rabbit IgG (Invitrogen, #A-11008; 1:400) secondary antibody, fluorophore-conjugated

phalloidin to visualize F-actin, and Alexa Fluor 647 goat anti-rat IgG (Invitrogen, #A-21247; 1:400) to detect the rat IgG1 portion of the surrogate Ag and thereby visualize Ag clusters. Coverslips were mounted onto slides using ProLong Diamond anti-fade reagent (Thermo Fisher, #P36965).

## Quantification of cluster-associated fluorescence and overlap

The total fluorescence intensity of clustered Ag or clustered signaling molecules at the B cell-APC interface was quantified as follows using custom MATLAB code (https://bitbucket.org/jscurll/bolger-munro_image_analysis_scripts/src/master/mask_fluorescence_img_LoG.m; *Scurll, 2019*). Each image was filtered (i.e. convolved) with a $3 \times 3$ averaging filter to smooth noise. Then, the standard deviation of the background pixel intensities in the filtered image was estimated to be $\sigma = 1.4826 \times \mathrm{MAD}(F_{3 \times 3}(I))$, where $\mathrm{MAD}(F_{3 \times 3}(I))$ is the median absolute deviation of the pixel intensities in the filtered image, $F_{3 \times 3}(I)$. A binary mask (Mask 1) was then defined by thresholding the pixel intensities in the filtered image at $N\sigma$ above the median intensity, where $N$=2 for pCD79, 2 for pSyk, and 3 for pCD19 experiments.

Next, the original, unfiltered image $I$ was filtered with Laplacian of Gaussian (LoG) filters of different widths $h$. Each LoG filter highlights fluorescent spots and edges of fluorescent features at a specific scale determined by $h$. Regions of approximately uniform pixel intensities become approximately zero after filtering with a LoG filter of width $h$ that is substantially smaller than the size of the uniform-intensity region. For each LoG-filtered image, $\mathrm{LoG}_h(I)$, the standard deviation $\sigma_h$ of the background pixel values was again estimated as described above, that is $\sigma_h = 1.4826 \times \mathrm{MAD}(\mathrm{LoG}_h(I))$. A second binary mask (Mask 2) was then defined by identifying pixels that exceeded a threshold $N\sigma_h$ in $\mathrm{LoG}_h(I)$ at any scale, where $N$=3 for pCD79, 5 for pSyk, or 4 for pCD19 experiments. Additionally, any pixel that passed this threshold was removed from the mask if its greatest negative response to the different LoG filters was greater than its maximum positive response. This last step helps to sharpen the edges of fluorescent features that could have been excessively blurred by filters of large width. A final binary mask was defined by taking the intersection of Mask 1 and Mask 2. Hence, pixels were retained in the final mask if and only if they were present in both Mask 1 and Mask 2.

To quantify fluorescence in an image, background fluorescence was estimated by calculating the median intensity of the pixels outside of the mask, and this was subsequently subtracted from the whole image. Next, the intensities of pixels outside of the mask were set to 0 to leave only those pixels that were present in the mask. Finally, these pixel intensities were summed to obtain the total fluorescence intensity in the image. For each B cell, the total fluorescence intensity derived in this manner for a given signaling molecule was normalized to the total fluorescence intensity of clustered Ag on the same B cell. To quantify overlap we used custom MATLAB code (https://bitbucket.org/jscurll/bolger-munro_image_analysis_scripts/src/master/quantify_fluorescence_overlap.m; *Scurll, 2019*) to calculate Manders' coefficients (*Manders et al., 1993*) after processing the images as described above to subtract background fluorescence and extract fluorescent features defined by the mask.

## Analysis of BCR signaling by immunoblotting

After being pre-treated for 1 hr with CK-666 or CK-689, $4 \times 10^6$ primary B cells in 0.1 ml imaging medium were added to anti-Ig$\kappa$-expressing COS-7 cells. Alternatively, $4 \times 10^6$ primary B cells in 0.1 ml of imaging medium were stimulated with 20 µg/ml soluble goat anti-mouse Ig$\kappa$ antibodies (Southern Biotech, #1050–01). Cells were lysed in RIPA buffer (30 mM Tris-HCl, pH 7.4, 150 mM NaCl, 1% Igepal (Sigma-Aldrich), 0.5% sodium deoxycholate, 0.1% SDS, 2 mM EDTA, 1 mM PMSF, 10 µg/ml leupeptin, 1 µg/ml aprotinin, 25 mM β-glycerophosphate, 1 µg/ml pepstatin A, 10 µg/ml soybean trypsin inhibitor, 1 mM Na$_3$MoO$_4$, 1 mM Na$_3$VO$_4$) and analyzed by immunoblotting. Filters were incubated overnight at 4°C with antibodies against Arp3 (Santa Cruz, #sc-15390; 1:1000), Arp2 (abcam, #ab128934; 1:1000), p34 (Millipore, #07–227; 1:1000), actin (Santa Cruz, #sc-47778; 1:5000), or CD79a (*Gold et al., 1991*; 1:5000), or with the following antibodies from Cell Signaling Technologies: pCD79a (#5173; 1:1000); pCD19 (#3571; 1:1000); CD19 (#3574; 1:1000); pERK (#9101; 1:1000), ERK (#9102; 1:1000), pAkt (#9271; 1:1000), or Akt (#9272; 1:1000). Immunoreactive bands were visualized using horseradish peroxidase-conjugated goat anti-rabbit IgG (Bio-Rad #170–6515; 1:3000), followed by ECL detection (GE Life Sciences). Blots were quantified and imaged using a Li-Cor C-DiGit imaging system. *Figure 9—figure supplement 6* shows complete images of blots from which sections are shown in *Figures 6*, *Figure 7*, and *Figure 8*.

## Single-particle tracking (SPT)

SPT using directly-conjugated Fab fragments of antibodies was performed as described previously (*Abraham et al., 2017*). Primary B cells from C57BL/6J mice were resuspended to $10^7$ cell/ml in RPMI-1640 without phenol red (Life Technologies #32404014) plus 5 mM HEPES. CK-666 or CK-689 was added to a final concentration of 100 µM and the cells were incubated for 1 hr at 37°C. The cells were then placed on ice and labeled for 5 min with 1 ng/ml Cy3-conjugated anti-mouse IgM Fab fragments (Jackson ImmunoResearch, #115-167-020), 100 ng/ml Cy3-conjugated Fab fragments from the 11–26c rat anti-mouse IgD monoclonal antibody (ATCC, #HB-250), or 10 ng/ml Cy3-conjugated Fab fragments of the 1D3 rat anti-mouse CD19 monoclonal antibody (ATCC, #HB-305). Fab fragments of the 11–26c antibody and 1D3 antibody were prepared by AbLab (Vancouver, Canada) using a Fab preparation kit (Pierce, #44985). The labeled B cells were then attached to coverslips that had been coated with 0.25 µg/cm$^2$ of the non-stimulatory M5/114 anti-MHC II monoclonal antibody (Millipore, #MABF33). SPT was carried out at 37°C using an Olympus total internal reflection fluorescence microscopy (TIRFM) system consisting of an inverted microscope (Olympus IX81) equipped with a 150X NA 1.45 TIRFM objective (Olympus), motorized filter wheel (Olympus), high-performance electorn multiplier (EM)-charge-coupled device (CCD) camera (Photometrics Evolve), and real-time data acquisition software (Metamorph). The 561 nm solid-state diode laser was used to excite Cy3-labeled samples. The TIRF plane was adjusted to yield a penetration depth of 85–90 nm from the coverslip and the cells were imaged for 10 s at 33 Hz.

Single particles were detected and tracked using Icy bioimaging analysis software (*de Chaumont et al., 2012*) with the UnDecimated Wavelet Transform Detector plug-in and the Multiple Hypothesis Spot-Tracking plug-in (*Chenouard et al., 2013*; *Chenouard et al., 2014*). Diffusion coefficients and confinement diameters were determined as described previously (*Abraham et al., 2017*). The diffusion model assumes that displacements arise from a two-dimensional Brownian diffusion process. Diffusion coefficients were calculated for individual tracks using a maximum likelihood estimation approach that accounts for noise due to positional errors in acquisition as well as blurring during acquisition (*Berglund, 2010*).

## APC-induced expression of the Nur77-GFP reporter gene, CD69, and CD86

COS-7 APCs expressing the anti-Igκ surrogate Ag ($2 \times 10^5$ cells per well) were cultured overnight in 12-well plates. Primary B cells from C57BL/6J mice or Nur77$^{GFP}$ mice were incubated in complete medium with 100 µM CK-666 or CK-689 for 1 hr and then $10^6$ B cells were added to each well containing APCs. After co-culturing the cells for 18 hr, the B cells and APCs were removed from the wells by pipetting. GFP levels were quantified by flow cytometry. To quantify cell surface expression of CD69 or CD86, the cells were resuspended in ice-cold FACS buffer (PBS with 2% FCS), Fc receptors were blocked by adding 25 µg/ml of the 2.4G2 monoclonal antibody (ATCC, #HB-197) for 5 min, and then the cells were stained for 30 min at 4°C with anti-CD69-FITC (eBioscience, #11-0691-82 1:200), anti-CD86-APC (eBioscience, #17-0862-82; 1:200), and DAPI (1:10,000) for 30 min. Fluorescence was quantified using an LSR II cytometer (Becton Dickinson) and the data were analyzed using FlowJo software (Treestar). Forward and side scatter, together with DAPI staining, were used to gate on single live (GFP-positive or DAPI-negative) B cells. The Ag-induced increase in fluorescence was calculated as the geometric mean for B cells cultured with anti-Igκ-expressing APCs, minus the geometric mean for B cells cultured with parental COS-7 cells that do not express the surrogate Ag. The gating strategies and sample calculations are shown in *Figure 9—figure supplement 1* (GFP expression) and *Figure 9—figure supplement 2* (CD69, CD86).

## CFSE proliferation and cell death assays

COS-7 APCs expressing the anti-Igκ surrogate Ag ($2 \times 10^5$ cells per well) were cultured overnight in 12-well plates. Primary B cells from C57BL/6J mice were resuspended in complete medium and labeled with 2 µM CFSE (Invitrogen, #C1157) for 8 min prior to being treated with CK-666 or CK-689 for 1 hr. The B cells were then added to the APCs and cultured in complete medium with 5 ng/ml BAFF (R&D Systems, #8876-BF-010) and 5 ng/ml IL-4 (R&D Systems, #404 ML-010). After 24–72 hr the cells were resuspended by pipetting, stained with 7-AAD (Thermo Fisher, #A1310; 1:1000) to identify dead cells, and analyzed by flow cytometry using an LSR II flow cytometer. Forward and side

scatter were used to gate on single B cells. CFSE fluorescence in live (7-AAD-negative) B cells was then quantified. FlowJo software was used to analyze the data.

## Statistical analysis

Two-tailed paired t-tests were used to compare mean values for matched sets of samples. The Mann-Whitney U test was used to compare ranks in samples with many cells and high variability (e.g. dot plots for immunofluorescence signaling data). For signaling experiments, outliers were identified using Robust Regression and Outlier Removal (ROUT) in GraphPad Prism with Q set to 1% (*Motulsky and Brown, 2006*). Kolmogorov-Smirnov tests were used to compare SPT cumulative frequency distributions and FRAP curves.

## Acknowledgements

We thank the UBC Life Sciences Institute Imaging Facility, the UBC Flow Cytometry Facility, Harshad Vishwasrao and Hari Shroff from the NIH Advanced Imaging and Microscopy (AIM) Resource, Linda Matsuuchi and Jia Wang for feedback on the manuscript, and Connor Morgan-Lang for help with data visualization.

## Additional information

### Funding

| Funder | Grant reference number | Author |
| --- | --- | --- |
| Canadian Institutes of Health Research | MOP-68865 | Michael R Gold |
| Natural Sciences and Engineering Research Council of Canada | RGPIN-2015-04611 | Daniel Coombs |
| Canadian Institutes of Health Research | PJT-152946 | Michael R Gold |

The funders had no role in study design, data collection and interpretation, or the decision to submit the work for publication.

### Author contributions

Madison Bolger-Munro, Conceptualization, Data curation, Formal analysis, Validation, Investigation, Visualization, Methodology, Writing—original draft, Project administration, Writing—review and editing, Designed and performed the majority of the experiments, Analyzed data, Wrote the manuscript; Kate Choi, Investigation, Writing—review and editing, Designed, performed, and analyzed experiments; Joshua M Scurll, Data curation, Software, Formal analysis, Validation, Methodology, Generated image analysis scripts, Performed data analysis; Libin Abraham, Conceptualization, Formal analysis, Investigation, Visualization, Designed, performed, and analyzed experiments; Rhys S Chappell, Software, Generated image analysis scripts, Performed data analysis; Duke Sheen, May Dang-Lawson, Investigation, Designed, generated, and validated essential reagents; Xufeng Wu, Investigation, Provided critical support for experimental set-up and data acquisition using the Zeiss Airyscan and instant structured illumination microscopes; John J Priatel, Resources, Provided the Nur77GFP mice, Commented on the manuscript; Daniel Coombs, Resources, Software, Funding acquisition, Methodology, Provided advice on quantitative image analysis; John A Hammer, Resources, Methodology, Project administration, Helped design experiments and interpret data; Michael R Gold, Conceptualization, Supervision, Funding acquisition, Validation, Visualization, Writing—original draft, Project administration, Writing—review and editing, Analyzed data, Wrote the manuscript

## Author ORCIDs

Madison Bolger-Munro (iD) https://orcid.org/0000-0002-8176-4824
Daniel Coombs (iD) https://orcid.org/0000-0002-8038-6278
Michael R Gold (iD) https://orcid.org/0000-0003-1222-3191

## Ethics

Animal experimentation: All of the animals were handled according to protocols approved by the University of British Columbia Animal Care Committee. Our animal protocols were approved by the University of British Columbia Animal Care Committee (mouse breeding license #A18-0334; animal use license #A15-0162). Mice were euthanized using halothane inhalation, followed by cervical dislocation, as detailed in our approved animal licenses.

## Decision letter and Author response

Decision letter https://doi.org/10.7554/eLife.44574.044
Author response https://doi.org/10.7554/eLife.44574.045

## Additional files

### Supplementary files

• Transparent reporting form
DOI: https://doi.org/10.7554/eLife.44574.040

### Data availability

Custom image analysis scripts are available online at https://github.com/madscience12/FIJImacros (copy archived at https://github.com/elifesciences-publications/FIJImacros) and https://bitbucket.org/jscurll/bolger-munro_image_analysis_scripts/src (copy archived at https://github.com/elifesciences-publications/Bolger-Munro_image_analysis_scripts).

The following dataset was generated:

| Author(s) | Year | Dataset title | Dataset URL | Database and Identifier |
|---|---|---|---|---|
| Bolger-Munro M, Choi K, Scurll JM, Abraham L, Chappell R, Sheen D, Dang-Lawson M, Wu X, Priatel JJ, Coombs D, Hammer JA, Gold MR | 2019 | Data from: Arp2/3 complex-driven spatial patterning of the BCR enhances immune synapse formation, BCR signaling and cell activation | https://dx.doi.org/10.5061/dryad.3bc215k | Dryad Digital Repository, 10.5061/dryad.j1fd7 |

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
