## [Decision Letter]

Thank you for submitting your article "Arp2/3 complex-driven spatial patterning of the BCR enhances immune synapse formation, BCR signaling and cell activation" for consideration by *eLife*. Your article has been reviewed by three peer reviewers, and the evaluation has been overseen by a Reviewing Editor and Arup Chakraborty as the Senior Editor. The following individuals involved in review of your submission have agreed to reveal their identity: Pavel Tolar (Reviewer #1); Bebhinn Treanor (Reviewer #2).

The reviewers have discussed the reviews with one another and the Reviewing Editor has drafted this decision to help you prepare a revised submission.

Summary:

This is a well done study that extends previous works on the organization of the B cell immune synapse and on the role of actin polymerization in BCR clustering, transport and signaling. The main conclusion is that actin polymerization specifically by the ARP2/3 complex drives movement and fusion of BCR/Ag clusters in B cells interacting with membrane antigen and that this boosts proximal BCR signaling as well as late events, such as B cell activation. These results are compatible with the previous idea that dynamic B cell lamellipodia are involved in grabbing antigen and that actin-based membrane shape, mechanics and flow promote BCR clustering and signaling.

The experimental system uses COS-7 cells expressing B cell antigen, which is good as this system mimics B cell-APC interactions better than artificial substrates. The COS-7 cells are paired with the established A20 B cell model, or in the signaling experiments with naive mouse B cells. The imaging and tools for visualisation of the immune synapse are cutting edge.

Using the CK666 inhibitor and siRNA, the authors report that actin polymerization through the ARP2/3 complex did not affect spreading of the B cells or the amount of the antigen gathered, but affected the movement and fusion of the antigen clusters. This is also illustrated by videos of the antigen flowing with the actin. SIM imaging of the B cells on glass substrates shows this clearly, as here the dramatic actin flow clearly depends on ARP2/3, similarly as it does in T cells.

The ARP2/3-inhibited cells show reduced proximal BCR signaling, particularly the early tyrosine phosphorylation of the BCR and of CD19, suggesting that the actin mediated movement and clustering is important in amplifying antigen-mediated B cell activation. Remarkably, the authors also see importance of this early ARP2/3 activity on late responses, such as expression of activation markers and proliferation, which are reduced in their intensity. Controls show that indeed this is through activity of ARP2/3 during the early contact of the B cells with the APCs. Inhibiting of ARP2/3 also leads to enhanced "tonic" signaling, similarly as inhibition of all actin polymerisation, suggesting that the ARP2/3-generated F-actin pool is the relevant structure regulating this aspect of BCR signaling.

Essential revisions:

The major concern with the otherwise excellent study is that the authors may overreach in some interpretations and there is some concern that the CK666 displays effects that go beyond the Arp3 siRNA, raising issues about interpretations based on CK666 using in this way (1 hr pretreat at 100 µM).

1) Video 1 (Figure 1B) shows the process of BCR-Ag clustering and SMAC formation over time. Clusters appear (t=0-1min), presumably upon B-cell contact, clusters move ~1µm centripetally where they coalesce into larger assemblies t=2-3 min), a few straggler clusters flow into the central array (t=3-5 min) and a SMAC assembly forms by ~10 min) although the forces acting on clusters and driving coalescence into a single SMAC structure are not obvious. The example illustrated in this video is unique. BCR-Ag formation and clustering in other cells shown in other videos do not exhibit all of these properties. Specifically, centripetal movement of BCR-Ag clusters is not commonly observed.

2) To assess BCR-Ag clustering and SMAC formation together with B-cell F-actin structures, the authors use F-tractin-GFP to visualize dynamic B-cell actin structures and BCR-Ag clustering after B-cells contact an APC (Video 2). BCR-Ag clusters first form upon initial contact of the B-cell and APC. These initial clusters increase in size as the B-cell spreads and extends across the APC surface via cycles of protrusion and retraction. Little centripetal movement of clusters occurs in this cell. Peripheral F-actin retrograde flow is not detectable in the time-lapse sequence (perhaps the time interval is too long to detect it). In this sequence, a single BCR-Ag microcluster was observed to move centripetally ~1µm before merging into an existing larger cluster (this movement was analyzed by kymography in Figure 1E). It is striking that as BCR-Ag clusters expand within the growing B-cell-APC contact zone, the regions become devoid of B-cell F-actin, a point that may be relevant to formation or stability of the emerging SMAC. There are no predominant F-actin structures flowing centripetally during formation of the SMAC. Indeed, BCR-Ag clusters expand in size and often coalesce with each other, but they do not appear to experience much directed retrograde flow; many simply expand in place or merge upon contact with neighboring clusters.

3) In video 3, the authors observe the behavior of B-cells spreading on a rigid glass IgG-coated coverslip using GFP-F-tractin probe. The authors describe B-cell behavior on glass as having long-lived membrane protrusions with no obvious membrane retraction events over the 10 min sequence. In contrast, the well-spread cell in video 3 exhibits dynamic membrane extension (lamellipodia and filopodia) and retraction around its circumference during the entire 10 min sequence. Moreover, one could quantify the rate of retrograde flow in this well-spread cell because the rate of time-lapse collection is sufficient to capture lamellipodial retrograde flow in some regions. The authors conclude that B-cells encountering antigen on an APC display complex and dynamic membrane probing and retraction behavior compared to when they are on glass. This appears to be the case, however, a comparison with B-cells spread on glass is comparing apples and oranges, especially when different rates of time-lapse collection are used. Nonetheless, it is important to describe observed cell behaviors and cluster dynamics as accurately as possible. To observe B-cells spreading on APCs may require different imaging modalities (faster image acquisition, additional focal planes) to appreciate its complex and intriguing behaviors.

4) Experiments aimed at blocking Arp2/3 complex activity implicated Arp2/3 complex in coalescence of BRC-Ag microclusters to form SMAC structures. The effects on BCR-Ag cluster dynamics were slightly different for Arp3 siRNA-mediated depletion compared to pharmacologic inhibition using CK-666; specifically, several examples of clusters experiencing centripetal flow are observed in the cell treated with Arp3 siRNA (Video 5). Additionally, in the cell treated with CK-666, the nature of cluster formation is quite different from that in cells depleted of Arp2/3 complex using siRNA; clusters are linear and appear to align and expand along filopodial-like protrusive structures. What accounts for this difference in formation and behavior of clusters under these two conditions?

5) Requirement of retrograde flow to form a SMAC. The authors do not provide compelling evidence in support of a role for Arp2/3-mediated retrograde flow in SMAC formation in B-cell-APC pairs. However, since they demonstrate that washout of CK-666 restored upregulation of CD69, a downstream effect of BCR-induced B cell activation (Figure 9—figure supplement 3), a question is whether or not a SMAC will form in B-cell-APC pairs upon washout of CK666. If the appropriate time-lapse imaging were used to simultaneously observe GFP-F-tractin in B-cells during washout of a CK-666 block, direct observations of F-actin retrograde flow and cluster flow during SMAC formation may be possible.

6) Finally, centripetal retrograde flow could be driven by a myosin-dependent process. This could be examined directly using inhibitors of myosin II. Disruption of global Arp2/3 complex actin networks is expected to exert perturbations on the whole cell cytoskeleton. Pinpointing the specific Arp2/3 dependent-activities during B-cell signaling will be challenging using this somewhat sledge-hammer approach.

Regarding essential experiments, as discussed above the videos are quite instructive in that the A20 cell synapse has a <1 µm rim of centripetal F-actin flow that transitions rapidly into < 1 µm myosin arc zone that leave the BCR microclusters in a loose assembly of ~1 µm BCR-Ag clusters/condensates scattered across a large area >5 µm in diameter. The final stage of "cSMAC" formation looks more like condensate fusion than transport. The tests for this would be different than for cytoskeletal transport. One way or the other, BCR consolidates in the center, but the F-actin appears to be permitting this rather than driving it. In the Arp3 KD the filopodia display robust centripetal F-actin flow as expected so the BCR clusters are transported similarly to the control siRNA, but there appear to be greater barriers to consolidation of the BCR clusters, so perhaps F-actin structures under these conditions are less permissive along the lines of work of Batista on F-actin constraining BCR movement. The CK666 seems to kill both the flow in the filopodia and create barriers to condensate fusion, so it is a very different picture that looks "poisoned" with the cell being paralysed in an unexpected way. The critical experiment that should be done is to use CK666 in the Arp3 KD to determine if CK666 has effects that go beyond the best Arp3 KD you can achieve in A20? If so, it could be because of incomplete KD, dominant effects of how CK666 inhibits Apr2/3, or off target effects perhaps also targeting cellular energy. Reversibility of CK666 effects would be good to see, but might not address the issues as any toxicity that makes the effects different than the Arp3 siRNA might also be reversed. In these experiments and controls, the authors could be asked to ensure that the frequency of imaging is optimised to best reveal proposed transport processes and to distinguish these from other non-correlated events. What is hoped is that this complete set of experiments (that completes the matrix of conditions set out in the study) will resolve if CK666 has activities that are beyond Arp3 siRNA (and thus could be off target) and are all of the events associated with cSMAC formation truly correlated with F-actin mediated transport or are their processes that are either independent or even hindered by some F-actin structures. Analysis of diffusive movement or random transport of the microclusters could aid identifying/discussing underlying mechanisms.

---

## [Author Response]

Essential revisions:The major concern with the otherwise excellent study is that the authors may overreach in some interpretations and there is some concern that the CK666 displays effects that go beyond the Arp3 siRNA, raising issues about interpretations based on CK666 using in this way (1 hr pretreat at 100 µM).1) Video 1 (Figure 1B) shows the process of BCR-Ag clustering and SMAC formation over time. Clusters appear (t=0-1min), presumably upon B-cell contact, clusters move ~1µm centripetally where they coalesce into larger assemblies t=2-3 min), a few straggler clusters flow into the central array (t=3-5 min) and a SMAC assembly forms by ~10 min) although the forces acting on clusters and driving coalescence into a single SMAC structure are not obvious. The example illustrated in this video is unique. BCR-Ag formation and clustering in other cells shown in other videos do not exhibit all of these properties. Specifically, centripetal movement of BCR-Ag clusters is not commonly observed.

We have addressed these comments in the following ways:

1) The reviewer is correct in that it is mainly the BCR microclusters that form at the periphery of the B cell-APC contact site, especially on newly-formed membrane protrusions, that move centripetally. To more accurately describe overall BCR microcluster movement, we now state “When A20 murine B-lymphoma cells expressing the HEL-specific D1.3 transgenic BCR (A20 D1.3 B cells) were added to the APCs, they rapidly formed BCR-Ag microclusters throughout the contact site. Microclusters that formed at the periphery moved centripetally. Closer to the center of the contact site, microclusters coalesced with each other such that a cSMAC developed within ~10 min (Figure 1B).”

2) We now show additional videos of microcluster dynamics in control cells (4 control siRNA-transfected cells in Video 4 and 4 CK-689-treated cells in Video 6). In these videos, a number of BCR microclusters are seen to move centripetally. To illustrate this, we now show additional kymographs of microclusters that exhibit centripetal movement in Figure 2—figure supplement 4G, H, including one in which two microclusters both move centripetally before coalescing. The reader is referred to these additional videos and kymographs in the third paragraph of the subsection “The Arp2/3 complex is required for centralization of BCR-Ag microclusters”.

3) For all of the cells (18-48 cells per condition) that had been previously quantified in Figure 2E, F and Figure 2—figure supplement 3), we now quantify the centripetal movement of BCR microclusters. To do this, we calculated for every frame in the video the distance between each microcluster and the center of mass of the Ag fluorescence (which at later times, this is the center of the cSMAC). The average value for all of the microclusters on a single cell was determined at each time point and was normalized to the maximum average distance for that cell. These new graphs (Figure 2—figure supplement 4E, F) show that, on average, net microcluster movement towards the center of the cell is minimal when the Arp2/3 complex is inhibited. The reader is referred to these additional graphs in the aforementioned paragraph.

With regard to the forces that promote the condensation of aggregated microclusters within the actin-depleted regions, this process may be driven by other cytoskeletal forces such as actomyosin contraction or microtubule-based mechanism. We note this in the Discussion (fifth paragraph).

2) To assess BCR-Ag clustering and SMAC formation together with B-cell F-actin structures, the authors use F-tractin-GFP to visualize dynamic B-cell actin structures and BCR-Ag clustering after B-cells contact an APC (Video 2). BCR-Ag clusters first form upon initial contact of the B-cell and APC. These initial clusters increase in size as the B-cell spreads and extends across the APC surface via cycles of protrusion and retraction. Little centripetal movement of clusters occurs in this cell. Peripheral F-actin retrograde flow is not detectable in the time-lapse sequence (perhaps the time interval is too long to detect it). In this sequence, a single BCR-Ag microcluster was observed to move centripetally ~1µm before merging into an existing larger cluster (this movement was analyzed by kymography in Figure 1E). It is striking that as BCR-Ag clusters expand within the growing B-cell-APC contact zone, the regions become devoid of B-cell F-actin, a point that may be relevant to formation or stability of the emerging SMAC. There are no predominant F-actin structures flowing centripetally during formation of the SMAC. Indeed, BCR-Ag clusters expand in size and often coalesce with each other, but they do not appear to experience much directed retrograde flow; many simply expand in place or merge upon contact with neighboring clusters.

We agree with the reviewer(s) and some of these points were addressed in our response to comment #1. We have inserted additional statements into the text that highlight the distinction between the centripetal movement of microclusters that form within actin-rich regions (especially those that form on membrane protrusions) and those that are closer to the center of the cell and which coalesce and condense in these actin-depleted regions (subsection “B cells generate dynamic actin-rich protrusions to scan for Ags on APCs”, second paragraph).

Moreover, we now show kymographs of additional microclusters exhibiting centripetal motion within actin-rich regions (Figure 1E). With regard to the kymograph shown in Figure 1E, we more clearly indicate that this refers specifically to microclusters that form in actin-rich regions and now state: “BCR-Ag microclusters formed within actin-rich regions, including newly formed protrusions Figure 1C, Video 2). […] As determined from the kymographs in Figure 1E, the velocities at which the peripheral F-actin structures and BCR-Ag microclusters moved towards the center of the synapse were similar, suggesting that actin dynamics drive the initial centripetal movement of these microclusters.”

The reviewer is correct that the peripheral F-actin flow is not obvious in the time-lapse series shown in Figure 1C because the time intervals are too large. However, they are clearly evident in Video 2, from which these still images are taken from, and in the kymographs in Figure 1E.

3) In video 3, the authors observe the behavior of B-cells spreading on a rigid glass IgG-coated coverslip using GFP-F-tractin probe. The authors describe B-cell behavior on glass as having long-lived membrane protrusions with no obvious membrane retraction events over the 10 min sequence. In contrast, the well-spread cell in video 3 exhibits dynamic membrane extension (lamellipodia and filopodia) and retraction around its circumference during the entire 10 min sequence. Moreover, one could quantify the rate of retrograde flow in this well-spread cell because the rate of time-lapse collection is sufficient to capture lamellipodial retrograde flow in some regions. The authors conclude that B-cells encountering antigen on an APC display complex and dynamic membrane probing and retraction behavior compared to when they are on glass. This appears to be the case, however, a comparison with B-cells spread on glass is comparing apples and oranges, especially when different rates of time-lapse collection are used. Nonetheless, it is important to describe observed cell behaviors and cluster dynamics as accurately as possible. To observe B-cells spreading on APCs may require different imaging modalities (faster image acquisition, additional focal planes) to appreciate its complex and intriguing behaviors.

We agree that the membrane dynamics of B cells spreading on immobilized anti-Ig versus an Ag-bearing APC are very different phenomena and that it is sufficient to describe the behaviors without belaboring the differences. Hence we replaced the section of the original manuscript, with what we feel is a more accurate description of the mode in which B cells spread on immobilized anti-Ig. Importantly, we have qualified our statement about how this differs from B cells probing the surface of an APC. We now say: “B cells spreading on immobilized anti-Ig formed multiple large lamellipodia, some of which persisted for up to 10 min (for example, Figure 1F [blue arrowheads]). They also formed short dynamic filopodia, many of which were retracted or engulfed by newly formed lamellipodia.”

With regard to quantifying the rate of actin retrograde flow in B cells spreading on immobilized anti-Ig, we did this in Figure 3E and Video 8 and conclude: “In control CK-689-treated cells (Figure 3E, Video 8), the F-actin networks were highly dynamic at the cell edge, with actin retrograde flow rates of ~2.5 μm/min”.

4) Experiments aimed at blocking Arp2/3 complex activity implicated Arp2/3 complex in coalescence of BRC-Ag microclusters to form SMAC structures. The effects on BCR-Ag cluster dynamics were slightly different for Arp3 siRNA-mediated depletion compared to pharmacologic inhibition using CK-666; specifically, several examples of clusters experiencing centripetal flow are observed in the cell treated with Arp3 siRNA (Video 5). Additionally, in the cell treated with CK-666, the nature of cluster formation is quite different from that in cells depleted of Arp2/3 complex using siRNA; clusters are linear and appear to align and expand along filopodial-like protrusive structures. What accounts for this difference in formation and behavior of clusters under these two conditions?Regarding essential experiments, as discussed above the videos are quite instructive in that the A20 cell synapse has a <1 µm rim of centripetal F-actin flow that transitions rapidly into < 1 µm myosin arc zone that leave the BCR microclusters in a loose assembly of ~1 µm BCR-Ag clusters/condensates scattered across a large area >5 µm in diameter. The final stage of "cSMAC" formation looks more like condensate fusion than transport. The tests for this would be different than for cytoskeletal transport. One way or the other, BCR consolidates in the center, but the F-actin appears to be permitting this rather than driving it. In the Arp3 KD the filopodia display robust centripetal F-actin flow as expected so the BCR clusters are transported similarly to the control siRNA, but there appear to be greater barriers to consolidation of the BCR clusters, so perhaps F-actin structures under these conditions are less permissive along the lines of work of Batista on F-actin constraining BCR movement. The CK666 seems to kill both the flow in the filopodia and create barriers to condensate fusion, so it is a very different picture that looks "poisoned" with the cell being paralysed in an unexpected way. The critical experiment that should be done is to use CK666 in the Arp3 KD to determine if CK666 has effects that go beyond the best Arp3 KD you can achieve in A20? If so, it could be because of incomplete KD, dominant effects of how CK666 inhibits Apr2/3, or off target effects perhaps also targeting cellular energy. Reversibility of CK666 effects would be good to see, but might not address the issues as any toxicity that makes the effects different than the Arp3 siRNA might also be reversed. In these experiments and controls, the authors could be asked to ensure that the frequency of imaging is optimised to best reveal proposed transport processes and to distinguish these from other non-correlated events. What is hoped is that this complete set of experiments (that completes the matrix of conditions set out in the study) will resolve if CK666 has activities that are beyond Arp3 siRNA (and thus could be off target) and are all of the events associated with cSMAC formation truly correlated with F-actin mediated transport or are their processes that are either independent or even hindered by some F-actin structures.

As shown in Figure 2—figure supplement 1, the Arp3 KD is not complete, so there is likely some residual Arp2/3 activity in the siRNA-transfected cells. We feel that this is the most likely explanation for why CK-666 has a more profound effect on actin and BCR microcluster dynamics/organization. We feel that the proposed experiment in which we would treat Arp3 siRNA-transfected cells with CK-666 would not distinguish between the drug inhibiting the activity of the residual Arp2/3 complexes in the KD cells from potential off-target effects of the drug. With regard to the comment that BCR microclusters in CK-666-treated and Arp3 siRNA-transfected cells exhibit different morphology and movement, we feel that this is largely attributable to the unique morphological changes and actin remodeling exhibited by different B cells when they interact with APCs that have complex plasma membrane organization. To address this, associated with Figure 2A, B we now show videos of BCR microcluster movement in 4 representative CK-689-treated, CK-666-treated, control siRNA-transfected, and Arp3 siRNA-transfected cells (Videos 4-7) instead of just one representative cell for each condition. Importantly, to extend these analyses to larger numbers of control and Arp2/3 complex-inhibited cells, we have quantified BCR microcluster movement and cSMAC formation in multiple ways. These quantitative analyses show that, when averaged over tens of cells, the inhibition of BCR microcluster dynamics is inhibited to a similar extent by CK-666 and Arp3 siRNA. This is true for average BCR microcluster area (Figure 2E, F; note that the y-axis scales on panels E and F are different), the number of BCR-Ag clusters per cell (Figure 2—figure supplement 4C, D), the average distance of BCR microclusters from the center of total Ag fluorescence (new data in Figure 2—figure supplement 4E, F), and the number of clusters required to contain 90% of the Ag fluorescence (Figure 2—figure supplement 4I, J). Moreover, we show that CK-666 and Arp3 siRNA inhibit cSMAC formation to the same extent. In these experiments, ~15% of the CK-666-treated and Arp3 siRNA-transfected cells are still able to form cSMACs (Figure 2A, B). Consistent with this, some microclusters do exhibit centripetal movement in the CK-666-treated and Arp3 siRNA-treated cells.

With regard to the possibility that the CK-666 is poisoning the cells or targeting cellular energy generation, BCR signaling in response to soluble Ags is completely normal in the presence of CK-666 (Figure 6D, Figure 6—figure supplement 2B, Figure 8—figure supplement 1B). Moreover, PMA-induced upregulation of CD69 and CD86 is unaffected by the presence of CK-666 for the entire 18 hr duration of this experiment (Figure 9E). Finally, we also used flow cytometry to directly show that treating primary B cells with CK-666 for up to 24 hr did not result in significant cell death (Figure 2—figure supplement 2A, B, referred to in the first paragraph of the subsection “The Arp2/3 complex is required for centralization of BCR-Ag microclusters”).

In our response to previous comments, we indicated that we agree with the reviewer and now describe more clearly that centripetal movement of microclusters occurs in actin-rich regions whereas within actin-depleted microclusters aggregate and condense. Specifically, we added the following sentences:

“[…] these data suggest that Arp2/3 complex-dependent actin structures encage BCR-Ag microclusters, such that the actin retrograde flow drives their initial centripetal movement. This appears to be required for the subsequent formation of a cSMAC, which occurs in the actin-depleted central region of the cell.”

The reviewer notes that microclusters within actin-depleted regions of the CK-666-treated and Arp3 siRNA-transfected cells appear not to coalesce to the same extent as those in actin-depleted regions of control cells. In the fifth paragraph of the Discussion, we discuss how microtubules may mediate the centripetal movement of BCR microclusters across the actin-depleted central region of the contact site and facilitate the final stages of cSMAC formation. In this paragraph we note that the formation of branched actin at the periphery of the immune synapse is required for capturing the plus ends of microtubules and for moving the microtubule network towards the Ag-presenting surface. Hence, Arp2/3 complex activity may be a prerequisite for the microtubule-based centralization that leads to cSMAC formation.

Analysis of diffusive movement or random transport of the microclusters could aid identifying/discussing underlying mechanisms.

Distinguishing random diffusion from directed super-diffusive motion would indeed be interesting but is not practical in our system because the B cells are also moving on the surface of the APC to some extent. Hence, this type moment scaling spectrum analysis is not feasible in this system.

5) Requirement of retrograde flow to form a SMAC. The authors do not provide compelling evidence in support of a role for Arp2/3-mediated retrograde flow in SMAC formation in B-cell-APC pairs.

We have endeavored to clarify this point and to be more precise about the requirement of Arp2/3 complex function versus retrograde flow in cSMAC formation (which we define as >90% of the total Ag fluorescence be contained in 1-2 large central clusters). In describing how inhibiting the Arp2/3 complex impairs cSMAC formation (Figure 2), we state: “The failure to form a cSMAC when the Arp2/3 complex is depleted or inhibited was due to decreased merger of individual BCR microclusters into larger clusters … Thus, BCR-mediated gathering of APC-bound Ags into a cSMAC at the center of the synapse is strongly dependent on the functions of the Arp2/3 complex.”

We go on to say in the Discussion that our data are consistent with Arp2/3 complex-dependent retrograde flow within actin-rich regions being an initial *requisite* step in cSMAC formation. The coalescence of BCR microclusters in the central actin-depleted regions of the cell may not be directly dependent on the Arp2/3 complex and likely does not involve actin retrograde flow. Specifically, in the first paragraph of the Discussion, we say that Arp2/3 complex activity, which generates retrograde actin flow, facilitates the initial centripetal movement of BCR microclusters and is required for cSMAC formation. We also say that “[…] inhibiting Arp2/3 complex activity may block a critical initial step in BCR-Ag microcluster centralization”.

We feel that these are precise and accurate statements as they implicate Arp2/3 complex function in cSMAC formation but do not imply that the coalescence of BCR microclusters into a cSMAC within the actin-depleted central region of the cell is dependent on actin retrograde flow.

However, since they demonstrate that washout of CK-666 restored upregulation of CD69, a downstream effect of BCR-induced B cell activation (Figure 9—figure supplement 3), a question is whether or not a SMAC will form in B-cell-APC pairs upon washout of CK666. If the appropriate time-lapse imaging were used to simultaneously observe GFP-F-tractin in B-cells during washout of a CK-666 block, direct observations of F-actin retrograde flow and cluster flow during SMAC formation may be possible.

Confirming that washing out the CK-666 drug restores BCR microcluster coalescence and cSMAC formation, and that this correlates with the restoration of CD69 upregulation (which had been shown in Figure 9—figure supplement 4 of the original manuscript) is an excellent suggestion. We have now carried out such an experiment using fixed cell imaging, which was sufficient to demonstrate that this is in fact the case. These new data are shown in Figure 9—figure supplement 4B-D. We now say “Once the drug was removed, the cells recovered their ability to aggregate BCR microclusters and form a cSMAC (Figure 9—figure supplement 4B-D) and B cell activation could proceed normally, consistent with the effects of CK-666 on actin dynamics being rapidly reversible (Yang et al., 2012).”

6) Finally, centripetal retrograde flow could be driven by a myosin-dependent process. This could be examined directly using inhibitors of myosin II. Disruption of global Arp2/3 complex actin networks is expected to exert perturbations on the whole cell cytoskeleton. Pinpointing the specific Arp2/3 dependent-activities during B-cell signaling will be challenging using this somewhat sledge-hammer approach.

We addressed this question by using pnBB to directly inhibit myosin and now show in Figure 9—figure supplement 5C, D that this does not prevent BCR microclusters from coalescing into a cSMAC, a process that we demonstrated is dependent on Arp2/3 complex activity and is associated with microcluster centripetal movement that is mediated by actin retrograde flow. This argues that the Arp2/3 complex does not act via a myosin-dependent process to drive centripetal movement of BCR microclusters.

However, the role of myosin in B cell responses to APC-bound Ags is an interesting question and we have now addressed this in Figure 9—figure supplement 5A, B. We found that treating primary B cells with pnBB caused a small decrease in the upregulation of CD69 or CD86, which was not statistically significant (i.e. p>0.05). This is in contrast to the Arp2/3 complex inhibitor CK-666, which caused significant reductions in these B cell activation responses. This indicates that the Arp2/3 complex has a more important role in APC-induced B cell activation than myosin. These new experiments are described in the sixth paragraph of the Results subsection “Arp2/3 complex activity is important for BCR-induced B cell activation responses”.